# Differential susceptibility of male and female germ cells to glucocorticoid-mediated signaling

Steven A Cincotta, Nainoa Richardson, Mariko H Foecke, Diana J Laird*

Department of Obstetrics, Gynecology and Reproductive Sciences, Center for Reproductive Sciences, Eli and Edythe Broad Center of Regeneration Medicine and Stem Cell Research, University of California, San Francisco, San Francisco, United States

*For correspondence:
Diana.laird@ucsf.edu

Competing interest: The authors declare that no competing interests exist.

**Abstract** While physiologic stress has long been known to impair mammalian reproductive capacity through hormonal dysregulation, mounting evidence now suggests that stress experienced prior to or during gestation may also negatively impact the health of future offspring. Rodent models of gestational physiologic stress can induce neurologic and behavioral changes that persist for up to three generations, suggesting that stress signals can induce lasting epigenetic changes in the germline. Treatment with glucocorticoid stress hormones is sufficient to recapitulate the transgenerational changes seen in physiologic stress models. These hormones are known to bind and activate the glucocorticoid receptor (GR), a ligand-inducible transcription factor, thus implicating GR-mediated signaling as a potential contributor to the transgenerational inheritance of stress-induced phenotypes. Here, we demonstrate dynamic spatiotemporal regulation of GR expression in the mouse germline, showing expression in the fetal oocyte as well as the perinatal and adult spermatogonia. Functionally, we find that fetal oocytes are intrinsically buffered against changes in GR signaling, as neither genetic deletion of GR nor GR agonism with dexamethasone altered the transcriptional landscape or the progression of fetal oocytes through meiosis. In contrast, our studies revealed that the male germline is susceptible to glucocorticoid-mediated signaling, specifically by regulating RNA splicing within the spermatogonia, although this does not abrogate fertility. Together, our work suggests a sexually dimorphic function for GR in the germline, and represents an important step towards understanding the mechanisms by which stress can modulate the transmission of genetic information through the germline.

## eLife assessment

This work reports a **valuable** finding on glucocorticoid signaling in male and female germ cells in mice, pointing out sexual dimorphism in transcriptomic responsiveness. The **convincing** evidence provided supports an inert GR signaling despite the presence of GR in the female germline and GR-mediated alternative splicing in response to dexamethasone treatment in the male germline. The work may interest basic researchers and physician-scientists working on reproduction and stress-related disease conditions.

## Introduction

The impact of stress on mammalian reproductive capacity has long been appreciated. The hypothesis that psychological stress not only limits the ability to conceive, but may also impact the health of offspring has gained considerable attention in recent years. A growing body of work has demonstrated

that rodent models of stress can induce a variety of neurologic and behavioral responses that may persist across generations, even in generations unexposed to the initial stressor (*Franklin et al., 2010*; *Wu et al., 2017*; *Gapp et al., 2014*; *Franklin et al., 2011*; *Weiss et al., 2011*; *Saavedra-Rodríguez and Feig, 2013*; *Babb et al., 2014*; *Manners et al., 2019*; *Bohacek et al., 2015*; *Gapp et al., 2017*; *Razoux et al., 2017*; *Chan et al., 2020*). The persistence of such responses for up to three generations suggests that stress signals can induce epigenetic changes in the germline that propagate. As the committed precursors of eggs and sperm established during early embryogenesis, germ cells are responsible for transmitting genetic as well as epigenetic information across generations. Perturbations of the messenger can potentially corrupt the transmission of this information, thus understanding the mechanism by which stress can lead to epigenetic alterations in the germline remains a crucial unanswered question.

It has been demonstrated that prenatal treatment of rodents with synthetic glucocorticoid stress hormones is sufficient to recapitulate the transgenerational changes seen in physiologic stress models; however, it remains unclear how glucocorticoids are sensed and remembered by developing germ cells (*Short et al., 2016*; *Moisiadis et al., 2017*; *Constantinof et al., 2019b*; *Constantinof et al., 2019a*). Multigenerational effects of preconception paternal exposures to stress were shown to rely on an indirect mechanism; glucocorticoid (cortisol) altered the small RNA cargo released in extracellular vesicles from epididymal epithelial cells to maturing sperm (*Chan et al., 2020*; *Short et al., 2016*). These hormones bind and activate the nuclear hormone receptor glucocorticoid receptor (GR; encoded by the *Nr3c1* gene), thus implicating GR-mediated signaling in transgenerational inheritance of stress. The engagement of GR with its ligand in the cytoplasm promotes translocation to the nucleus where it then binds DNA response elements and recruits transcriptional cofactors to modulate target gene expression. As such, GR functions as a ligand-inducible transcription factor, allowing for robust changes to gene expression in response to stress hormones. While GR is known to be a potent transcriptional regulator in a wide variety of cell types (*Whirledge and DeFranco, 2018*), its function and targets in the cells of the developing gonad remain less clear.

Abundant levels of GR have been identified in the Leydig and peritubular myoid cells of the testis (*Stalker et al., 1989*; *Schultz et al., 1993*; *Biagini and Pich, 2002*; *Weber et al., 2000*; *Hazra et al., 2014*), which supports the possibility that stress is sensed by somatic cells, which in turn elicit epigenetic changes in germ cells. Prior evidence for GR expression in the germ cells is sparse. In the adult mouse testis, one study suggested GR expression was limited to primary spermatocytes (*Schultz et al., 1993*), while others suggested that GR is expressed in both spermatogonia as well as spermatocytes (*Biagini and Pich, 2002*; *Weber et al., 2000*). In humans, multiple studies have shown robust GR expression in differentiating spermatogonia, as well as low expression in primary spermatocytes (*Nordkap et al., 2017*; *Welter et al., 2020*). In the female, a single study of the human fetal ovary found evidence for GR expression by IHC in the oocytes of a 9-week-old embryo (*Poulain et al., 2012*), although expression was heterogeneous across the tissue. A more recent study of mouse fetal oocyte development using single-cell RNA-sequencing revealed increasing expression of GR in the oocytes between E12.5 and E14.5 (*Ge et al., 2021*). Together these data suggest that GR may be expressed in the rodent / human germline, but in limited developmental windows. However, no studies to date have directly assessed the cell-intrinsic role of GR in the germ cells. While several publications describe a variety of responses of the male and female germline to glucocorticoid treatment (*Short et al., 2016*; *Poulain et al., 2012*; *Orazizadeh et al., 2010*; *Ren et al., 2021*; *Mancini et al., 1966*; *Yazawa et al., 2000*; *Mahmoud et al., 2009*; *Gapp et al., 2021*; *González et al., 2010*; *Tohei et al., 2000*; *Wei et al., 2019*; *Yuan et al., 2020*; *Yuan et al., 2016*), it is unclear whether such changes arise from direct action of glucocorticoids on germ cells, or the indirect effects of systemic glucocorticoid treatment on hormone production by gonadal somatic cells and/or the hormone producing cells of the hypothalamus and pituitary.

Here, we characterized the expression of GR in the germline of the developing and adult mouse gonads, and determined the role of GR in normal germline formation and function. We discovered that GR is expressed in the female germline exclusively during fetal development, but absent from the adult oocyte. We demonstrated that the female germline is, surprisingly, resistant to changes in GR signaling. Negligible changes in transcription or meiotic progression of fetal oocytes resulted from either genetic deletion or agonism of GR with dexamethasone (dex), suggesting that the female germline is intrinsically buffered from changes in GR signaling. In contrast, we found that GR is expressed

in pro-spermatogonia of the perinatal testis and spermatogonia of the adult testis, and transcriptomic analysis of germ cells from dex-treated males revealed a potential role of GR in regulating RNA splicing. Together our data confirm the dynamic spatiotemporal regulation of GR expression in both the male and female germ cells, and suggest a sexually dimorphic role for this receptor in the mammalian germline.

## Results

### Spatiotemporal expression of GR and novel isoforms in fetal oocytes

To assess GR expression in the developing female germline, we performed immunofluorescence (IF) on mouse fetal ovaries ranging from embryonic day (E) 12.5 through E18.5. Female germ cells, identified by either *Pou5f1*-GFP (*Pou5f1-ΔPE-eGFP* transgene) or TRA98, showed robust levels of GR starting at E13.5, which waned as development progressed (*Figure 1A*). GR consistently localized to the nucleus of fetal germ cells, irrespective of developmental stage, suggesting that GR may function as a transcription factor within the female germline (*Figure 1B*). To rigorously quantify GR expression dynamics in germ cells, we normalized GR expression levels to DAPI signal at the individual cell level, using the membrane marker wheat germ agglutinin (WGA) for segmentation (*Figure 1—figure supplement 1A*). These studies confirmed that GR expression in the germline decreased from E13.5 to E18.5 (*Figure 1C*).

At postnatal day 0 (PN0), IF revealed a small number of oocytes at the cortex of the ovary with nuclear GR, with expression declining and virtually absent by PN2 (apart from sporadic cortical oocytes with cytoplasmic GR; *Figure 1—figure supplement 1B*). GR was not expressed in postnatal oocytes between PN5 and PN21, nor in adult oocytes of all follicular stages (*Figure 1—figure supplement 1C*). In the somatic compartment, GR was virtually absent from FOXL2$^+$ granulosa cells, but was strongly expressed in the theca cell layer (marked by α-SMA *McKey et al., 2020*) from as early as PN7 through adulthood (*Figure 1—figure supplement 1D*).

It has been hypothesized that spatiotemporal differences in GR expression across different tissues can be due to the use of alternative, non-coding *Nr3c1* exon 1 splice isoforms in different cell types (*Turner et al., 2006*). To explore this possibility, we looked for evidence of exon 1 alternative splicing in our paired-end RNA-seq data of E15.5 sorted *Pou5f1*-GFP$^+$ germ cells and GFP$^-$ somatic cells (see *Figure 2D* for details). Sashimi plots visualizing the junctions between exon 2 and known exon 1 variants demonstrated that while somatic cells exclusively use exon 1B, 1D, and 1 F, female germ cells use a much wider range of known exon 1 variants, including 1 A, 1B, 1 C, 1D, and 1 F (*Figure 1D*). Surprisingly, we also found evidence of three novel splice isoforms exclusively in the germ cells, labeled here as predicted exons 1α, 1β, and 1γ (*Figure 1D*, blue lines). These cell-type-specific patterns of exon 1 isoform usage were validated using RT-PCR on bulk and sorted populations, with primers designed specifically to amplify each of the known and predicted exon junctions (*Figure 1E*). This wider diversity of *Nr3c1* exon 1 variants in the female germline suggests that alternative promoters may be cooperatively regulating the expression of GR, thus leading to the more dynamic temporal regulation of GR seen in the germline in comparison to the soma.

### Genetic deletion of GR leads to minimal changes in the female germline

The strong, nuclear localization of GR in germ cells of the fetal ovary led us to hypothesize that GR may be functioning as a transcriptional regulator of early oocyte development. This possibility was raised in a single-cell RNA-seq study of the developing mouse ovary that noted that GR expression is highly correlated with progression of fetal oocytes through prophase I (*Ge et al., 2021*). In zebrafish, genetic ablation of GR led to accelerated ovarian aging and decline in fertility later in life (*Maradonna et al., 2020*; *Faught et al., 2020*; *Facchinello et al., 2017*). To interrogate the potential role of GR in meiosis, we utilized a mouse line with a constitutive genetic deletion of GR. *Nr3c1* exon 3 floxed mice were crossed to a constitutive β-actin Cre mouse line to generate a constitutive null allele of GR (henceforth referred to as 'KO'). *Nr3c1* exon 3 deletion has previously been shown to result in efficient loss of functional GR protein (reviewed extensively elsewhere *Whirledge and DeFranco, 2018*), which we confirmed by IF at E17.5 (*Figure 2A*), immunoblot at E13.5 (*Figure 2B*), and qRT-PCR at E15.5 (*Figure 2C*). To functionally test whether meiotic progression was delayed or disrupted following loss of GR, we scored meiotic prophase I in spreads for E15.5 WT and KO ovaries stained for SYCP3,

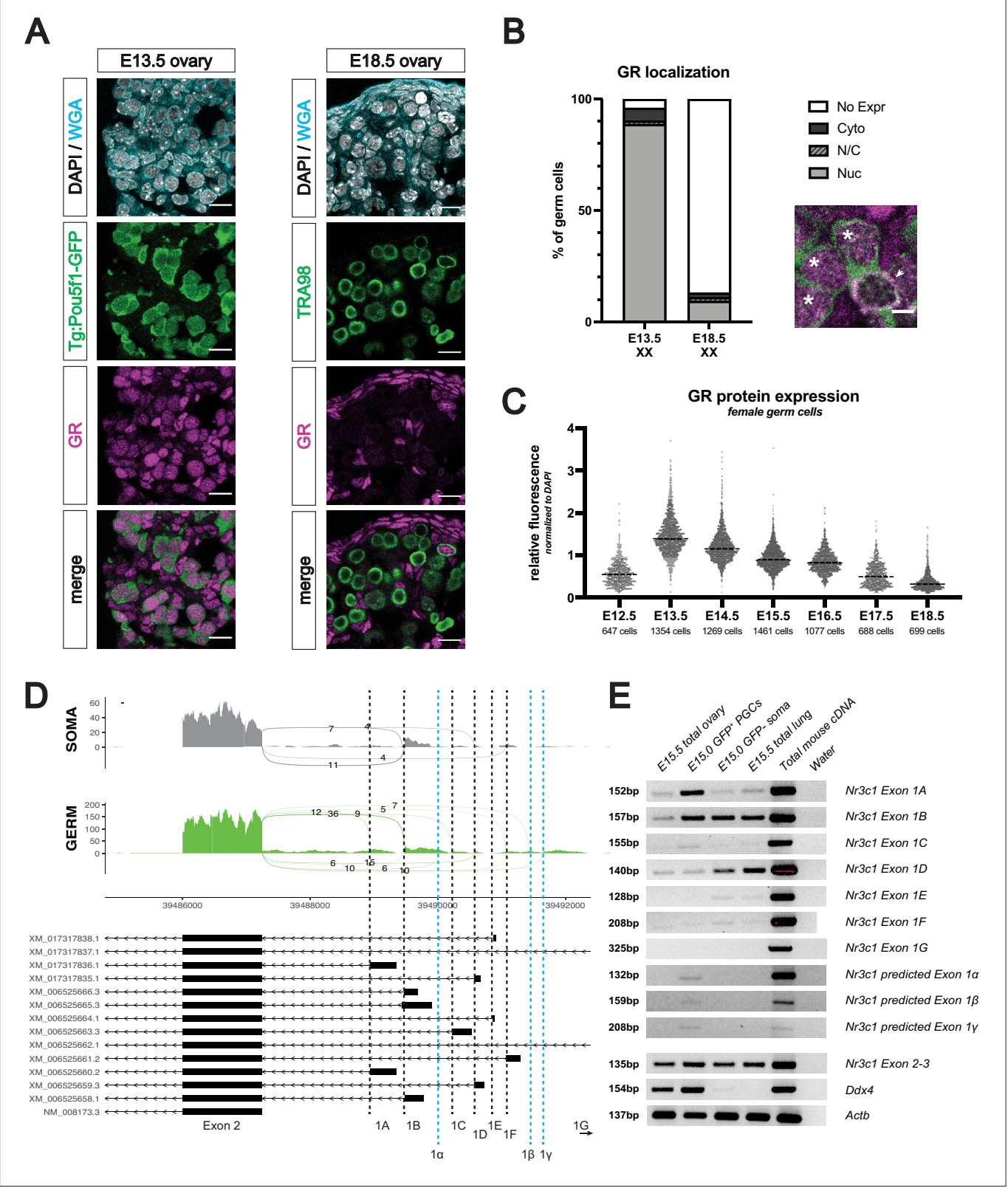

**Figure 1.** The glucocorticoid receptor is expressed in the developing fetal oocyte. (**A**) IF staining showing expression of GR in mouse fetal ovary sections at E13.5 (left) and E18.5 (right), counterstained with DAPI. Germ cells are marked by either transgenic *Pou5f1*-GFP or TRA98. Cellular membranes were stained with wheat germ agglutinin (WGA) to facilitate computational segmentation of individual cells. Scale bars: 15 μm. (**B**) Quantification of GR subcellular localization within germ cells of the fetal ovary. Cells from the ovaries of three individual embryos were scored,

*Figure 1 continued on next page*

*Figure 1 continued*

with a total of 1423 cells and 859 cells analyzed at E13.5 and E18.5, respectively. Zoomed in image showing examples of both nuclear (asterisk) and cytoplasmic (arrowhead) GR staining in germ cells. Scale bar: 5 μm. (**C**) Quantitative IF analysis of relative GR protein expression in the germ cells across developmental time. Individual cells were computationally segmented using WGA, and GR protein levels were normalized to DAPI on an individual cell basis. Images and total cell numbers counted were obtained from a minimum of three ovaries from three individual embryos at each developmental stage. (**D**) Sashimi plots showing differences in alternative exon 1 splicing events at the *Nr3c1* locus between ovarian germ and somatic cells. Plots were generated from paired-end RNA-seq data of E15.5 germ and somatic cells (saline control; ***Figure 3A***). Previously annotated exon 1 variants have been arbitrarily labeled as exons 1 A through 1 G (with exon 1 A being closest to exon 2). Three novel exon 1 splice sites identified in this study have been labeled as predicted exons 1α, 1β, and 1γ (marked by the dotted blue lines). (**E**) RT-PCR validation of exon 1 variant usage in bulk E15.5 ovary, as well as sorted populations of germ and somatic cells at E15.0. Total mouse cDNA and water serve as positive and negative controls for all reactions, respectively. E15.5 total lung lysate serves as a positive control for GR expression. Primer set spanning *Nr3c1* exon 2–3 junction (present in all isoforms) was used as a positive control for total *Nr3c1* transcript.

The online version of this article includes the following source data and figure supplement(s) for figure 1:

**Source data 1.** Original blot file (1 of 3) for RT-PCR in ***Figure 1E*** (*Nr3c1* exons 1 A, 1B, 1 C, 1D, 1E and 1 F).

**Source data 2.** Original blot file (2 of 3) for RT-PCR in ***Figure 1E*** (*Nr3c1* exons 1 G, *Nr3c1* exon 2–3, and *Ddx4*).

**Source data 3.** Original blot file (3 of 3) for RT-PCR in ***Figure 1E*** (*Nr3c1* exons 1α, 1β, 1γ, and *Actb*).

**Source data 4.** PDF file containing ***Figure 1E*** with original RT-PCR blot files, highlighting bands used in ***Figure 1E*** with sample annotations.

**Figure supplement 1.** The glucocorticoid receptor is not expressed in the late postnatal and adult oocyte.

SYCP1, and γH2AX (***Figure 2D***). No significant changes in substage distribution could be detected between WT and KO nuclei, suggesting that loss of GR does not disrupt meiotic progression. qRT-PCR on E15.5 WT and KO bulk ovary tissue for a panel of genes known to have important roles in meiotic progression similarly revealed no changes in the transcription of meiotic genes (***Figure 2E***).

To assess whether GR regulates the transcriptional landscape of alternative cellular pathways in the female germline, we employed two orthogonal transcriptomic approaches. We first performed low-input RNA-seq on FACS-sorted *Pou5f1*-GFP⁺ germ cells from individual WT and KO ovaries at E17.5. While our differential gene expression analysis pipeline revealed robust expression differences between WT female and WT male germ cells (***Figure 2F and i***), we saw no statistically significant differentially expressed genes between female WT and KO germ cells (***Figure 2F and ii***). Conditional deletion of GR specifically in the germ cells using *Pou5f1-CreERT2* yielded the same result (***Figure 2F and iii***). In tandem, we performed single-cell RNA-seq on E15.5 WT and KO germ cells. All expected cell types of the fetal ovary were detected in both WT and KO embryos (***Figure 2—figure supplement 1A***), and total *Nr3c1* transcript was depleted in all cell types of the ovary (***Figure 2—figure supplement 1B***). Differential gene expression analysis of WT and KO cells within the germ cell cluster revealed an extremely low number of differentially expressed genes with minor fold changes (***Figure 2—figure supplement 1C***), corroborating our bulk RNA-seq results that the loss of GR has little effect on the transcriptional landscape of fetal oocytes.

## Developing ovarian somatic cells, but not germ cells, show GR transcriptional activation following exogenous glucocorticoid treatment

While homeostatic GR signaling does not appear to regulate fetal oocyte gene expression or meiotic progression, we next asked whether the GR agonist dexamethasone (dex) could elicit a transcriptional response in the female germline. Pregnant dams were administered either 10 μg dex / g or saline vehicle by intraperitoneal (IP) injection daily from E12.5 to E15.5, coinciding with the window of highest GR expression in the female germline. We validated this dosing regimen by sufficiency to induce expression of the canonical GR response gene *Fkbp5* in bulk ovary, testis, and lung tissue of E15.5 embryos (***Figure 3—figure supplement 1A***), confirming dex was able to transit the placenta to the embryos. Accordingly, we performed bulk RNA-seq on sorted *Pou5f1*-GFP⁺ germ cells and total GFP⁻ somatic cells from E15.5 ovaries of dex vs saline-treated dams. While the somatic cells showed a robust transcriptional response following dex treatment (1477 total differential genes with adjusted p-value ≤0.05), by comparison, the response was severely dampened in germ cells (156 total differential genes with adjusted p-value ≤0.05; ***Figure 3A***). GO term analysis of differentially expressed genes from the somatic cells showed low fold enrichment across a wide

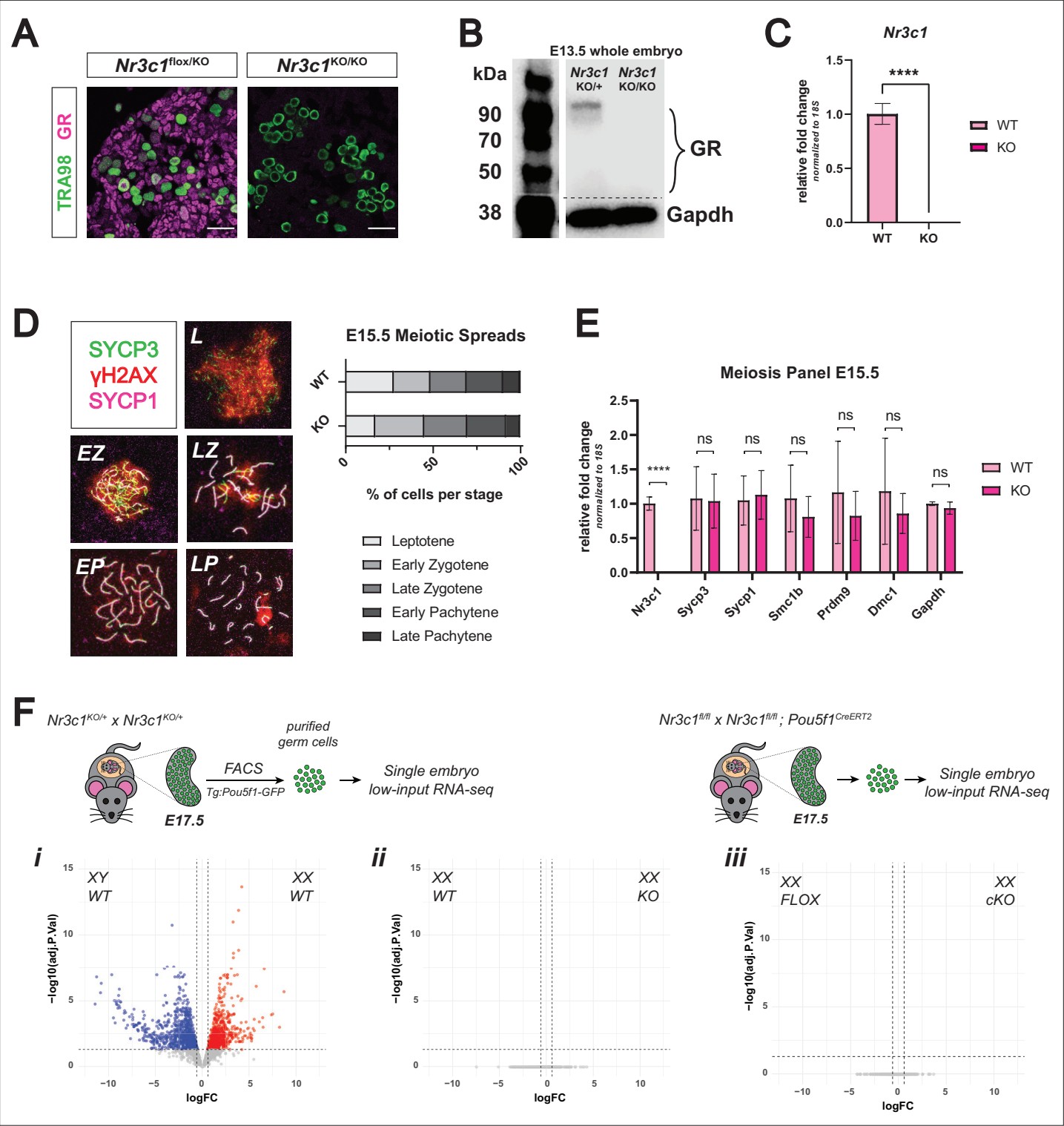

**Figure 2.** Genetic deletion of the glucocorticoid receptor leads to minimal changes in fetal oocytes. (**A**) IF staining for GR in E17.5 ovaries. *Nr3c1*flox/KO ovaries, which contain one functional floxed allele of GR show robust GR expression, whereas *Nr3c1*KO/KO ovaries homozygous for the deletion allele show complete loss of GR. Scale bars: 30 µm. (**B**) Western blot performed on whole cell lysate prepared from entire E13.5 embryos of different genotypes. Membranes were blotted with a GR antibody that recognizes all known GR isoforms, as well as GAPDH as a loading control.(**C**) qRT-PCR on bulk E15.5 WT (n=3) and KO (n=5) ovaries for *Nr3c1*, normalized to 18 S ribosomal RNA housekeeping gene using 2^ΔΔCt quantification method. Data are mean ±s.d., and p-values were calculated for each gene using a two-tailed, unpaired t-test, where ****: p≤0.0001. (**D**) Meiotic spreads performed on germ cell nuclei from E15.5 WT and GR KO ovaries, Left: representative images of meiotic prophase I staging of spreads co-stained with SYCP3

*Figure 2 continued on next page*

*Figure 2 continued*

(green), SYCP1 (magenta), and γH2AX (red). Right: Quantification of relative substages based on manual scoring. For WT spreads, a total of 590 nuclei from five embryos were counted; For GR KO spreads, a total of 817 nuclei from seven embryos were counted. L: Leptotene; EZ: Early Zygotene; LZ: Late Zygotene; EP: Early Pachytene; LP: Late Pachytene. (**E**) qRT-PCR on bulk E15.5 WT (n=3) and KO (n=5) ovaries for a panel of meiotic genes, normalized to 18 S ribosomal RNA housekeeping gene using $2^{-\Delta\Delta Ct}$ quantification method. *Nr3c1* serves as a positive control to confirm complete GR knockout, and *Gapdh* serves as an unchanged negative control. Data are mean ±s.d., and p-values were calculated for each gene using a two-tailed, unpaired t-test, where ****: p≤0.0001, n.s.: not significant. (**F**) Bulk RNA-seq performed on Tg:*Pou5f1*-GFP⁺ sorted germ cells from E17.5 gonads. For each genotype, three to four single-embryo biological replicates were used for low-input library prep followed by 3′ Tag-Seq. Volcano plots show differentially expressed genes (logFC ≥0.6; adjusted p-value ≤0.05) between: (i) WT male and WT female germ cells, (ii) WT and GR KO female germ cells, and (iii) *Pou5f1*-CreERT2⁺ GR conditional knockout (cKO) female germ cells and floxed Cre-negative controls.

The online version of this article includes the following source data and figure supplement(s) for figure 2:

**Source data 1.** Original blot file for western blot in *Figure 2B* (anti-GR and anti-GAPDH).

**Source data 2.** PDF files containing *Figure 2B*, highlighting bands used in *Figure 2B* with sample annotations.

**Figure supplement 1.** scRNA-seq confirms genetic deletion of the glucocorticoid receptor leads to minimal changes in fetal oocytes.

range of unrelated categories, likely due to the heterogeneity of the GFP⁻ population (*Figure 3— figure supplement 1B*). The set of upregulated (i.e. dex-induced) genes in the germ cells did not show any significant GO term enrichment (*Figure 3—figure supplement 1B*). The few downregulated germ cell genes showed an enrichment for ribosomal assembly and translation (*Figure 3— figure supplement 1B*), although we did not detect significant changes in protein levels of any germ cell marker assessed. In somatic gonad cells, we verified that the dex-induced transcriptional changes led to changes at the protein level using an ex vivo culture system. Fetal ovaries dissected at E14.5 and cultured in hormone-depleted medium for 48 hr with and without 1 µM dex confirmed upregulation of PLZF (*Zbtb16*, the most highly upregulated gene in the somatic cells by RNA-seq) specifically in the Tg:Pou5f1-GFP⁻ soma and not the GFP⁺ germ cells (*Figure 3B and i*). Utilizing the same culture system, we analyzed meiotic spreads and confirmed that dex treatment does not alter meiotic prophase, in line with the lack of transcriptional response (*Figure 3—figure supplement 1C*).

## GR in fetal oocytes relocalizes in response to ligand, but resistance to glucocorticoid signaling cannot be explained by the presence of inhibitory isoforms

The observed lack of a transcriptional or phenotypic change in the germ cells to both deletion of GR and dex treatment in contrast to adjacent somatic cells led us to hypothesize that the female germline may be resistant to GR signaling. As the function of GR as a transcription factor requires nuclear localization, we tested whether its subcellular localization is altered in response to ligand. In ovaries cultured ex vivo without dex, GR localized to the cytoplasm of virtually all germ cells; in contrast, dex-treated cultures retained nuclear localization similar to fetal oocytes in vivo (*Figure 3B, ii*). As GR localization still dynamically responded to the presence or absence of ligand, despite not robustly altering transcription, an alternative modification to the GR protein specifically in the germ cells could be accounting for this attenuated activity.

GR is a highly modified protein, with a wide range of transcriptional isoforms, translational isoforms, and post-translational modifications (PTMs) described to date (*Weikum et al., 2017*). Multiple translation initiation sites in the *Nr3c1* gene lead to a variety of protein isoforms with varying truncations in the regulatory N-terminal domain of the protein (*Lu and Cidlowski, 2005*). The most truncated forms of GR (the 'GR-D' isoforms) can exert an inhibitory dominant negative effect (*Lu and Cidlowski, 2005*). We used an antibody designed specifically to recognize all known GR isoforms to immunoblot E13.5 ovary lysate but did not detect any GR-D isoforms (~50 kDa or less), confirming that this dominant negative isoform is not the cause of decreased GR activity in the germline (*Figure 3C*). In mice, alternative splicing and subsequent retention of *Nr3c1* intron 8 results in a GRβ isoform that functions as a dominant negative regulator of the predominant GRα isoform (*Hinds et al., 2010*). However, we saw no evidence of intron 8 retention in our paired-end RNA-seq reads (*Figure 3D*), suggesting that the GRβ isoform is also not the cause of attenuated GR activity in fetal oocytes.

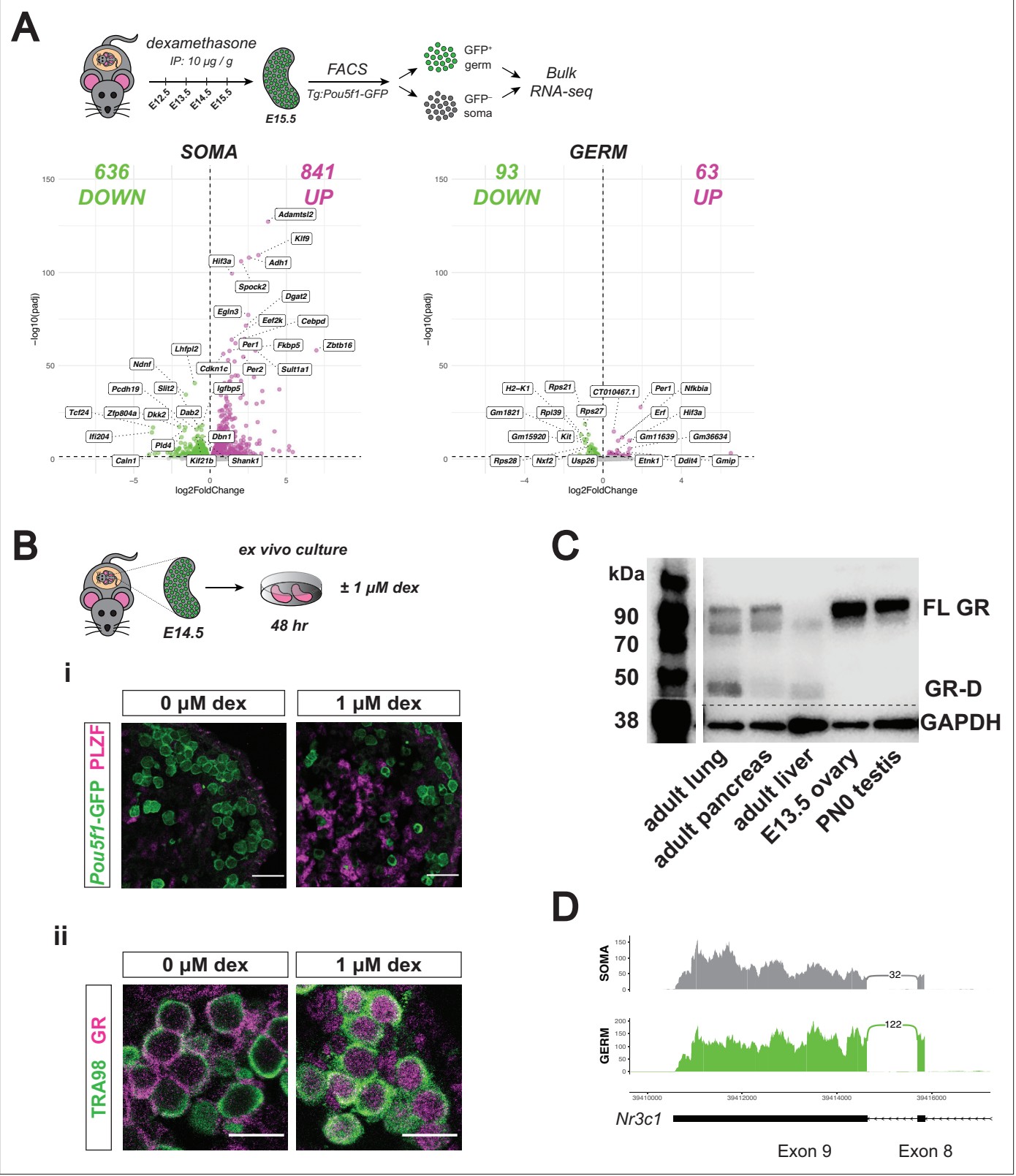

**Figure 3.** Developing ovarian somatic cells, but not germ cells, show robust GR transcriptional activation following exogenous glucocorticoid treatment. (**A**) Bulk RNA-seq performed on sorted Tg:Pou5f1-GFP⁺ germ cells and GFP⁻ somatic cells from fetal ovaries dosed in vivo with dex. Pregnant dams were injected via IP with either 10 µg / g dex or a saline vehicle control at E12.5, E13.5, E14.5, and E15.5, and ovaries were collected for sorting at E15.5. Three biological replicates were used per condition, each consisting of sorted cells from ovaries pooled together from a minimum of two entire

*Figure 3 continued on next page*

*Figure 3 continued*

independently dosed litters. Volcano plots show dex-induced differentially expressed genes (adjusted p-value ≤0.05) either upregulated (magenta) or downregulated (green) in comparison to vehicle controls for both germ cells, as well as total somatic cell population. (**B**) Ex vivo culture of E14.5 ovaries for 48 hr with and without 1 μM dex. (i) IF staining for PLZF (*Zbtb16*) showing induction specifically in the Tg:*Pou5f1*-GFP⁻ soma, scale bars: 30 μm. (ii) GR shows dynamic subcellular localization in response to ligand, scale bars: 15 μm. (**C**) Western blot performed on whole cell lysate prepared from adult lung, adult pancreas, adult liver, E13.5 whole ovary, and PN0 whole testis. Membranes were blotted with a GR antibody that recognizes all known GR isoforms, as well as GAPDH as a loading control. The full-length GR protein ("FL GR") can be seen at just above 90 kDa, and while the truncated inhibitory isoform of GR ("GR-D") can be seen just below 50 kDa. (**D**) Sashimi plots showing lack of evidence for intron 8 retention, which would lead to the inhibitory GRβ transcriptional isoform in either ovarian germ or somatic cells. Plots were generated from paired-end RNA-seq data of E15.5 germ and somatic cells (saline control; Figure 3A).

The online version of this article includes the following source data and figure supplement(s) for figure 3:

**Source data 1.** Original blot file for western blot in *Figure 3C* (anti-GR and anti-GAPDH).

**Source data 2.** PDF files containing *Figure 3C*, highlighting bands used in *Figure 3C* with sample annotations.

**Figure supplement 1.** In vivo dex dosing regimen validation, GO term analysis of RNA-seq from female germ cells dosed with dex in vivo, and meiotic spreads of dex-dosed female germ cell nuclei.

## Spatiotemporal expression of GR in the male germline

We next characterized the expression pattern of GR in the testis by IF from E12.5 through E18.5. In stark contrast to the female, germ cells of the fetal testis (marked by Tg:*Pou5f1*-GFP) showed no GR expression during early sex differentiation (*Figure 4A*). It was not until E17.5 that male germ cells began to express GR, and by E18.5, over 90% of all germ cells harbored GR in the nucleus (*Figure 4A and B*). We employed a similar quantitative imaging analysis as in the female to confirm that GR expression peaked in the male at E18.5 (*Figure 4C*).

As GR levels in prospermatogonia peaked at the end of fetal development, we next asked whether expression was maintained into postnatal development. Staining between PN0 (not shown) and PN2 revealed that strong nuclear GR expression was maintained in the TRA98⁺ germ cells shortly after birth (*Figure 4—figure supplement 1A*). At PN1, we also observed a wider diversity of *Nr3c1* exon 1 variants in the germ cells when compared to the soma (*Figure 4—figure supplement 1C*), consistent with observations in fetal oocytes. Staining at PN7, PN10, and PN14 revealed that GR expression was maintained in both PLZF⁺ and c-KIT⁺ spermatogonia, although at lower levels than at PN2 (*Figure 4—figure supplement 1A and B*). As anticipated based on prior reports (*Stalker et al., 1989*; *Schultz et al., 1993*; *Biagini and Pich, 2002*; *Weber et al., 2000*; *Hazra et al., 2014*), GR expression appeared high in the surrounding peritubular myoid cells and Leydig cells of the interstitium at all timepoints observed. By PN21, GR became highly restricted to the undifferentiated (PLZF⁺) and differentiating (c-KIT⁺) spermatogonia, yet was absent from more mature spermatocytes or spermatids (*Figure 4—figure supplement 1A and B*). Furthermore, this spermatogonia-restricted expression pattern was maintained into adulthood (*Figure 4D*), with minimal to no expression in spermatocytes (SYCP3⁺) and spermatids (Lectin⁺), suggesting a potential stage specific role for GR in the male germline.

## Germ cells of the perinatal testis show GR transcriptional regulation following exogenous glucocorticoid treatment

As fetal oocytes appeared to be resistant to both loss and overactivation of GR signaling, we next asked whether the male germline was similar. We performed bulk RNA-seq on sorted Pou5f1-GFP⁺ germ cells and GFP⁻ somatic cells isolated from PN1 testes after administering dex at E17.5, E18.5 and PN0 (as outlined in *Figure 5A* and Materials and methods section). In contrast to the female, differential expression analysis revealed a pronounced transcriptional response in male germ cells following dex treatment, within the same order of magnitude as the GFP⁻ soma (*Figure 5A*). GO term analysis on differentially expressed genes in the germ cells following dex treatment revealed a strong enrichment for genes related to the regulation of mRNA splicing (*Figure 5—figure supplement 1A*). The regulation of RNA splicing in germline cells is crucial for the proper progression of meiosis and spermatogenesis (*Kuroda et al., 2000*; *Li et al., 2007*; *O'Bryan et al., 2013*; *Schmid et al., 2013*; *Zagore et al., 2015*; *Naro et al., 2017*; *Liu et al., 2017*; *Xu et al., 2017*; *Horiuchi et al., 2018*; *Ehrmann et al., 2019*; *Legrand et al., 2019*; *Yuan et al., 2021*; *Wu et al., 2021*), and is conserved across species (*Mattox and Baker, 1991*; *Wu et al., 2016*; *Chen et al., 2019*). Downregulated genes

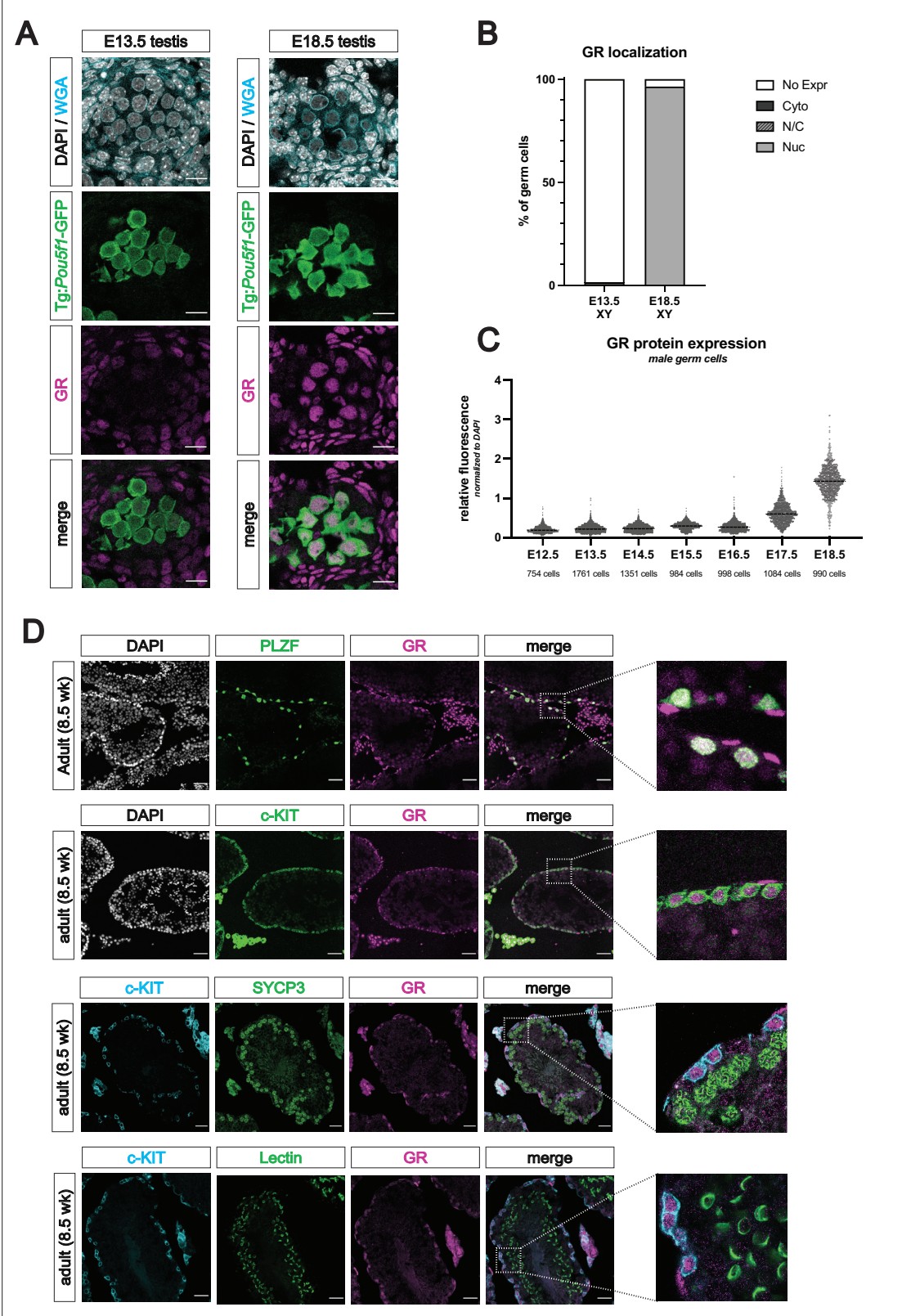

**Figure 4.** The glucocorticoid receptor is expressed in the perinatal prospermatogonia and the adult spermatogonia. (**A**) IF staining showing expression of GR in mouse fetal testis sections at E13.5 (left) and E18.5 (right), counterstained with DAPI. Germ cells are marked by transgenic *Pou5f1*-GFP. Cellular membranes were stained with wheat germ agglutinin (WGA) to facilitate computational segmentation of individual cells. Scale bars: 15 µm. (**B**) Quantification of GR subcellular localization within germ cells of the fetal testis. Cells from the testes of three individual embryos were scored

*Figure 4 continued on next page*

*Figure 4 continued*

manually, with a total of 2393 cells and 1364 cells analyzed at E13.5 and E18.5, respectively. (**C**) Quantitative IF analysis of relative GR protein expression across developmental time in germ cells. Individual cells were computationally segmented using WGA, and GR protein levels were normalized to DAPI on an individual cell basis. Images and total cell numbers counted were obtained from a minimum of three testes from three independent embryos at each developmental stage. (**D**) IF staining showing expression of GR in mouse adult testis sections. GR expression overlaps with PLZF⁺ undifferentiated spermatogonia (first row) and c-KIT⁺ differentiating spermatogonia (second row), zoomed images to highlight overlapping expression. GR is low to absent in SYCP3⁺ spermatocytes (third row) and PNA Lectin stained spermatids (fourth row). Scale bars: 50 µm (first two rows); 30 µm (last two rows).

The online version of this article includes the following figure supplement(s) for figure 4:

**Figure supplement 1.** The glucocorticoid receptor is expressed in the postnatal prospermatogonia.

also showed an enrichment for mitotic cell cycle progression, which is of interest at this PN1 time point given that the mitotically-arrested male germ cells will begin to re-enter mitosis at approximately PN2 (***Vergouwen et al., 1991***; ***Drumond et al., 2011***; ***Nagano et al., 2000***). GO analysis of differentially expressed genes in treated somatic cells showed a wider variety of terms, which is likely due to the heterogeneity of GFP⁻ cells collected. Downregulated genes were broadly enriched for extracellular matrix organization and cellular adhesion, whereas upregulated genes showed enrichment for canonical glucocorticoid response genes.

## Regulation of mRNA splicing in the early postnatal testis

Given the crucial role of transcript splicing in spermatogenesis, we first validated this potential link between GR and splicing. To our knowledge, no prior studies have implicated GR in the regulation of mRNA splicing in any cell type to date. To quantify any dose-dependent changes in splice factor gene expression, we performed qRT-PCR on bulk PN2 testis tissue from mice treated with three different doses of dex (0, 1, and 10 µg / g). Both *Tra2b* (a regulator of exon inclusion/skipping known to be expressed in the testis ***Grellscheid et al., 2011***) and *Srsf7* (a member of the SR-rich family of pre-mRNA splicing factors) showed a significant and dose-dependent decrease in expression in response to dex, confirming our RNA-seq results (***Figure 5B***). To test whether this dex-dependent decrease in a subset of splicing factors resulted in any changes in transcript isoforms within the germ cells, we utilized rMATS (***Shen et al., 2012***) to specifically quantify differential splicing events from our paired-end RNA-seq data. Sequencing libraries were constructed from full-length polyA transcripts that were reverse transcribed using random hexamer priming, thus allowing for sequencing coverage along the full-length transcript as well as detection of differential splicing events. This analysis revealed 63 splicing events that were significantly altered in response to dex (***Figure 5C***), with the vast majority categorized as skipped exon events. Together, these results confirmed a dex-dependent decrease in splice factor expression that may lead to exon skipping events in the male germline.

## Conditional deletion of GR in the male germline does not impact fertility

Given this link between GR signaling and splicing and the known role of transcript splicing in regulating spermatogenesis (***Kuroda et al., 2000***; ***Li et al., 2007***; ***O'Bryan et al., 2013***; ***Schmid et al., 2013***; ***Zagore et al., 2015***; ***Naro et al., 2017***; ***Liu et al., 2017***; ***Xu et al., 2017***; ***Horiuchi et al., 2018***; ***Ehrmann et al., 2019***; ***Legrand et al., 2019***; ***Yuan et al., 2021***; ***Wu et al., 2021***), we next sought to assess whether loss of GR would impact male fertility. Because full body deletion of GR results in lethality at birth due to defects in lung maturation (***Cole et al., 1995***), we generated a conditional deletion of GR in the germline using a transgenic *Prdm1*-Cre line (***Ohinata et al., 2005***). *Nr3c1*^flox/flox females were bred to *Nr3c1*^KO/+ males harboring the *Prdm1*-Cre transgene to obtain conditional knockout (cKO) pups with the genotype *Nr3c1*^KO/flox; *Prdm1*-Cre⁺, (where *Nr3c1*^+/flox; *Prdm1*-Cre^neg served as wildtype controls). IF staining at PN30 confirmed clear germ cell-specific loss of GR in the c-KIT⁺ prospermatogonia (***Figure 6A***), as well as the PLZF⁺ prospermatogonia (***Figure 6—figure supplement 1A***). To assess the fertility of GR cKO mice, 6-week-old males were crossed to wildtype females, and the presence of vaginal plugs and litter size were monitored for 8 weeks. All genotypes were able to mate and form vaginal plugs normally (data not shown). Control mice containing the *Prdm1*-Cre transgene, but functionally wild-type for GR (*Nr3c1*^flox/+) showed a significant decrease in the number of sired pups (***Figure 6B***), consistent with previously reported subfertility with this allele (***Mikedis and Downs, 2017***). GR cKO males were still fertile and produced viable pups (***Figure 6B***).

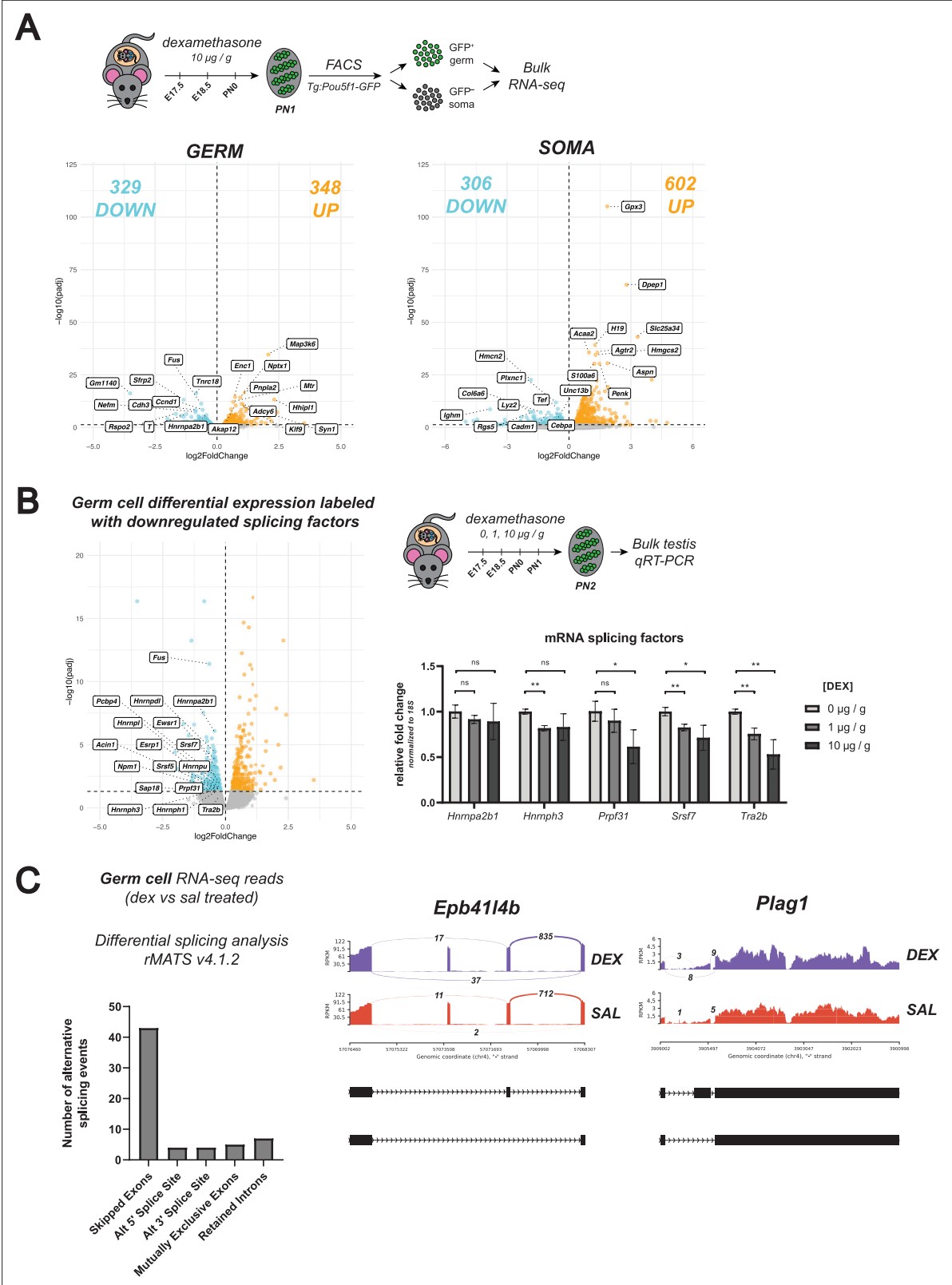

**Figure 5.** Glucocorticoid receptor signaling regulates the expression of RNA splicing factors in prospermatogonia. (**A**) Bulk RNA-seq performed on sorted Tg:Pou5f1-GFP⁺ germ cells and GFP⁻ somatic cells from postnatal testes dosed in vivo with dex. Pregnant dams were injected via IP with either 10 µg / g dex or a saline vehicle control at E17.5 and E18.5. Pups were then dosed with either dex or saline via subcutaneous injection at PN0, and testes were collected for sorting at PN1. Three biological replicates were used per condition, each consisting of sorted cells from testes pooled together from

*Figure 5 continued on next page*

*Figure 5 continued*

a minimum of two entire independently dosed litters. Volcano plots show dex-induced differentially expressed genes (adjusted p-value ≤0.05) either upregulated (orange) or downregulated (cyan) in comparison to vehicle controls for both germ cells, as well as total somatic cell population. (**B**) (Left) Plotting of germ cell differential expression analysis in (**A**), labeled specifically with downregulated splicing factors. (Right) qRT-PCR performed on bulk PN2 testis tissue from mice dosed with dex. Pregnant dams were injected via IP with either 1 µg /g dex, 10 µg / g dex, or a saline vehicle control at E17.5 and E18.5, and then pups were dosed with the equivalent condition via subcutaneous injection at PN0 and PN1. Genes queried were for RNA splicing factors found to be differentially downregulated in germ cells in the RNA-seq data (Figure 5A). Data are mean ±s.d., normalized to 18 S ribosomal RNA housekeeping gene using $2^{-\Delta\Delta Ct}$ quantification method, and p-values were calculated for each dose comparison using a two-tailed, unpaired t-test, where *: p≤0.05; **: p≤0.01; n.s.: not significant. (**C**) Differential transcript splicing analysis performed on sequencing reads derived from PN1 dex-dosed germ cells (Figure 5A). Paired-end RNA-seq results were analyzed using rMATS to detect significant differences in alternative splicing events between saline- and dex-treated germ cells. (Left) Bar graph showing raw numbers of significant (FDR ≤0.05) alternative splicing events, broken down by category. (Right) Representative sashimi plots highlighting select examples of 'skipped exons' events that are differentially regulated in germ cells in response to dex treatment.

The online version of this article includes the following figure supplement(s) for figure 5:

**Figure supplement 1.** GO term analysis of RNA-seq from male germ cells dosed with dex in vivo.

Although GR cKO males sired significantly less pups than WT controls, this was not significantly different from the effect produced by the presence of the *Prdm1*-Cre allele alone (*Figure 6B*). We therefore conclude that conditional deletion of GR in the germline does not further exacerbate the subfertility phenotype caused by the *Prdm1*-Cre allele, and ultimately yields viable progeny.

## Discussion

The growing evidence that responses to stress can persist across multiple generations has sparked considerable interest in understanding how the germline senses and conveys physiologic stress, and whether that occurs directly or indirectly. Our efforts to characterize the role of GR-mediated signaling in male and female germ cells represent an important step in understanding the intrinsic effects of stress hormones on the germline. In this study, we confirmed that germ cells of the testis and the ovary both show dynamic temporal regulation of GR expression over fetal and adult development, with distinct sex-specific differences. We demonstrated that in the context of pharmacologic activation or genetic deletion of GR, fetal oocytes show minimal transcriptional changes and no significant changes in progression through meiotic prophase I, suggesting that the fetal oocytes are somehow resistant to changes in GR signaling. In contrast, we show that prospermatogonia downregulate genes important for RNA splicing in response to GR activation, suggesting a potential role for stress signaling in regulating transcript diversity in the male germline. Together, our work demonstrates a sexually dimorphic response to GR signaling within the germline, and suggests alternative mechanisms by which the male and female germline may be differentially affected by stress.

### Glucocorticoid receptor immunofluorescence in the ovary and testis: discrepancies and similarities with the literature

Our time course of GR expression in the fetal ovary revealed a peak in GR expression in E13.5 fetal oocytes (*Figure 1C*), a result highly consistent with a scRNA-seq study that found GR was expressed in the developing oocytes (at both the transcript and protein level) between E12.5 to E14.5 (*Ge et al., 2021*). To our knowledge, there is no published evidence of GR protein in the adult rodent oocyte to date. We were unable to detect any appreciable levels of GR by IF in the adult oocytes of any follicular stage (*Figure 1—figure supplement 1C*), which contrasts with prior GR staining in caprine ovaries that showed expression in primordial and antral oocytes (*Pontes et al., 2019*). It is unclear whether this discrepancy is due to genuine species-specific differences, or differences in sample preparation or antibodies used. In humans, while GR has been detected in fetal oocytes at gestational week 9.2 (*Poulain et al., 2012*), the expression of GR in the adult ovary remains an open question.

Spatiotemporal localization of GR in the male germline here, however, is consistent with prior studies. In the somatic compartment, we observed strong and reproducible staining for GR in the peritubular myoid cells and interstitial Leydig cells of the fetal, postnatal and adult testis (*Figure 4A, D, Figure 4—figure supplement 1A and B*), consistent with previous studies (*Stalker et al., 1989*; *Schultz et al., 1993*; *Biagini and Pich, 2002*; *Weber et al., 2000*; *Hazra et al., 2014*). In the late

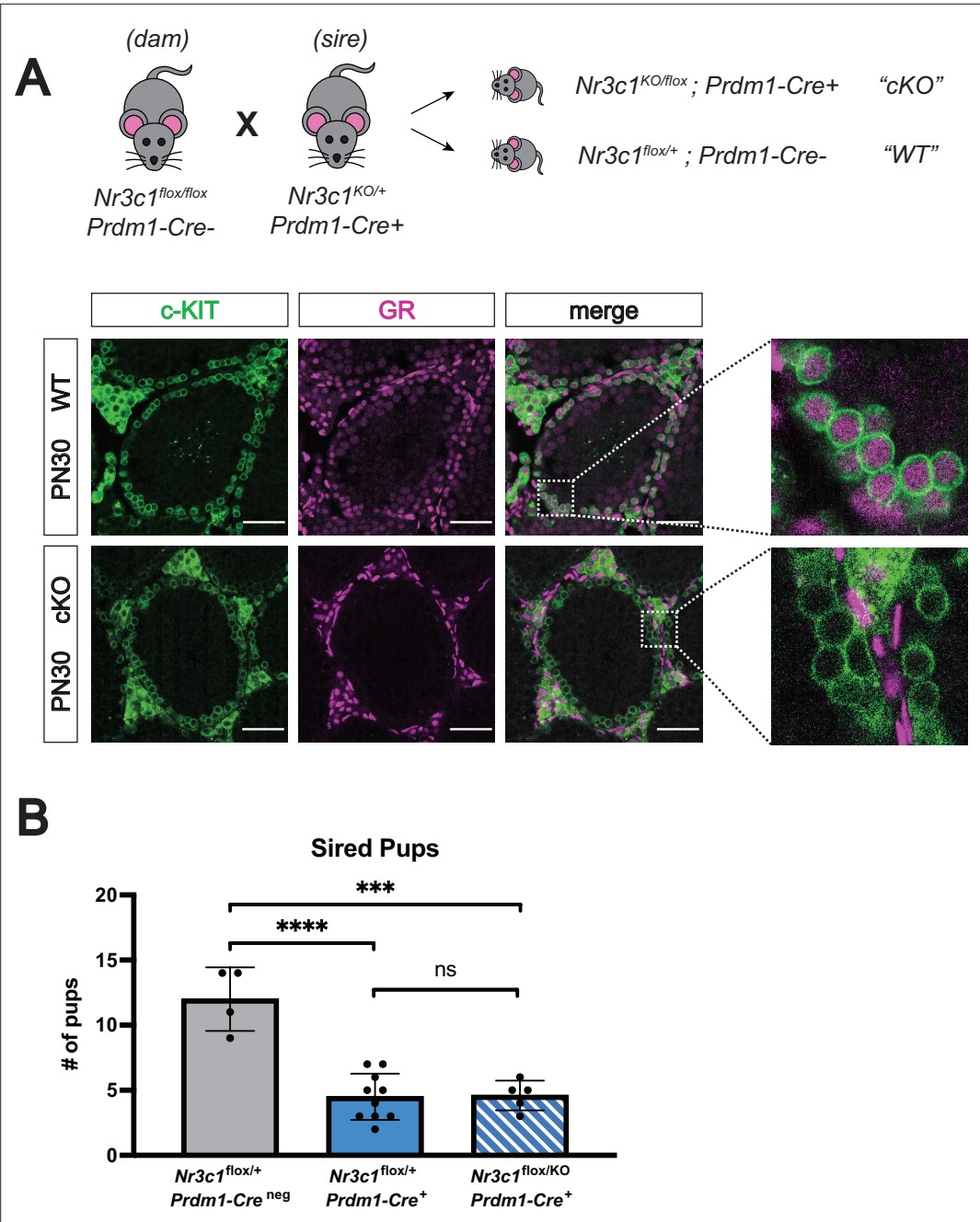

**Figure 6.** Conditional deletion of GR in the male germline does not affect fertility. (**A**) Validation of *Prdm1*-Cre mediated GR conditional knockout model by IF staining of PN30 testes, showing specific loss of GR in c-KIT+ spermatogonia. Genotypes are represented as cKO (*Nr3c1*KO/flox; *Prdm1*-Cre+) and WT (*Nr3c1*flox/+; *Prdm1*-Creneg). Scale bars: 50 µm. (**B**) Fertility test of GR conditional knockout males and controls crossed to WT females, comparing number of pups sired per litter by genotype. Data are mean ±s.d., and p-values were calculated between groups using a two-tailed, unpaired t-test, where ****: p≤0.0001, ***: p≤0.001 and n.s.: not significant.

The online version of this article includes the following figure supplement(s) for figure 6:

**Figure supplement 1.** Validation of *Prdm1*-Cre mediated GR conditional knockout.

postnatal and adult testis, we observed a highly specific pattern of GR expression in the germline, where it became restricted to undifferentiated (PLZF+) spermatogonia and differentiating (c-KIT+) spermatogonia, with little to no expression in spermatocytes (SYCP3+) or spermatids (Lectin+) (*Figure 4D*). This result is consistent with multiple rodent and human studies showing staining in the spermatogonia (*Biagini and Pich, 2002*; *Weber et al., 2000*; *Nordkap et al., 2017*; *Welter et al., 2020*).

## Discovery of novel germ cell-specific exon 1 isoforms in the 5' UTR of *Nr3c1*

Previous studies attribute the dynamic spatiotemporal regulation of GR across different cell types to the use of alternative promoters and exon 1 variants in the 5' UTR of *Nr3c1* (*Turner et al., 2006*; *McCormick et al., 2000*). In both the ovary and the testis, we observed that *Nr3c1* transcripts produced by the germ cells use a wider combination of exon 1 variants (both previously annotated, and novel germ-cell-specific variants) compared to their corresponding somatic cells (*Figure 1D*, *Figure 4—figure supplement 1C*), which was surprising given the cellular heterogeneity of the GFP⁻ somatic cell fractions. This suggests that a larger diversity of upstream regulatory DNA sequences may be available for regulating GR expression more dynamically in the germline compared to the soma. We also observed that GR is expressed just prior to the onset of meiotic initiation in male and female germ cells. This suggests GR may have an upstream regulatory element that is either regulated by a meiotic transcription factor, or that is sensitive to changes in the chromatin landscape induced in germ cells upon meiotic initiation.

## The female germline is insulated from changes in GR-mediated signaling

Given that GR is a potent transcription factor with a wide variety of functions across cell types (*Whirledge and DeFranco, 2018*), it was striking to find that both complete genetic deletion of GR and robust pharmacologic activation led to minimal changes in the transcriptional landscape of the female germline (*Figures 2F and 3A*, *Figure 2—figure supplement 1C*). Furthermore, the robust response to dex administration between E12.5 and E15.5 observed in ovarian soma in contrast to the meager response of adjacent germ cells (*Figure 3A*) suggested that the transcriptional response to changes in GR signaling was buffered in the cellular context of fetal oogonia. This is an interesting premise, as it may be advantageous for the germ cells to evolve protection against rampant GR-induced transcriptional changes in response to stress hormones experienced during gestation. Why the female germline has retained temporally restricted expression of GR given its apparent lack of a functional role, however, remains unclear. We have shown there is no evidence for inhibitory isoforms of GR, including both known truncated protein isoforms, as well as the GRβ transcriptional isoform (*Figure 3C and D*). We did, however, observe that GR is able to change its subcellular localization in response to ligand (*Figure 3B, ii*), suggesting that the ligand binding and nuclear translocation abilities of GR remain intact despite this attenuated function in female germ cells.

A possible mechanism for insulating fetal germ cells from response to GR signaling involves PTMs to the receptor that could interfere with its ability to bind DNA, interact with other transcription factors, or recruit other cofactors required for its transcriptional activity. While a wide variety of PTMs to GR have been documented (*Weikum et al., 2017*), the limiting number of germ cells that can be obtained from the fetal ovary has made the detection of these PTMs via biochemical methods difficult. One particular PTM of interest is acetylation of GR by the circadian histone acetyltransferase protein, CLOCK. Acetylation of lysines within the hinge region of GR by CLOCK has been shown to result in normal ligand binding and nuclear translocation, but a complete loss of the ability of GR to both activate and repress transcription (*Nader et al., 2009*), very similar to what is seen in fetal oocytes. Interestingly, our single-cell RNA-seq data shows a strong enrichment for *Clock* expression in the germ cells in comparison to the soma (data not shown), further bolstering the idea that CLOCK could acetylate and inactivate GR specifically in the germ cells. Further biochemical analyses will be required to determine whether GR is being modified in this manner, or by any other PTMs. Lastly, while our RNA-seq results show a very minimal response to GR signaling in germ cells at the transcriptional level, we cannot rule out a potential non-canonical signaling function of GR, such as the regulation of cellular kinases (reviewed extensively elsewhere *Oakley and Cidlowski, 2013*), or the regulation of non-coding RNAs at the post-transcriptional level.

## GR in the regulation of RNA splicing in the male germline

In contrast to the lack of dex-induced transcriptional changes in the female germ cells, the male germline showed a considerable transcriptional response to dex treatment at PN1 (*Figure 5A*). While the somatic cells of the testis showed a more stereotyped glucocorticoid response with induction of canonical GR response genes such as *Tsc22d3*, *Klf9*, and *Per1*, germ cells did not show changes in these canonical genes. Instead, the downregulated transcripts showed a strong enrichment for genes involved in RNA processing and RNA splicing (*Figure 5—figure supplement 1A*). We validated that GR suppression of splice protein expression is dex dose-dependent (*Figure 5B*), and found that treatment with dex led to an increase in exon skipping events at specific loci across the genome (*Figure 5C*). To our knowledge, this is the first time GR has been implicated in regulating the expression of genes involved in RNA splicing. Interestingly, it has been well documented in the literature that a full genetic deletion of a single splice factor leads to impairments in spermatogenesis, and ultimately infertility (*Kuroda et al., 2000*; *Li et al., 2007*; *O'Bryan et al., 2013*; *Zagore et al., 2015*; *Liu et al., 2017*; *Xu et al., 2017*; *Horiuchi et al., 2018*; *Ehrmann et al., 2019*; *Legrand et al., 2019*; *Yuan et al., 2021*; *Wu et al., 2021*). We suspect that dex treatment leads to fewer differential splicing events than a full splice factor deletion, given that dex treatment causes a broader decrease in splice factor expression without entirely abolishing any single splice factor. Unfortunately, we have been unable to assess the consequences of dex-induced splicing decreases on fertility, as dex- and/or glucocorticoid-mediated inhibition of testosterone production in the Leydig cells would impair spermatogenesis (*Ren et al., 2021*; *Bambino and Hsueh, 1981*; *Hales and Payne, 1989*; *Kavitha et al., 2006*; *Orr and Mann, 1992*; *Orr and Mann, 1990*; *Smith and Walker, 2014*; *Rengarajan and Balasubramanian, 2007*; *Rengarajan and Balasubramanian, 2008*; *Saez et al., 1977*; *Sankar et al., 2000*; *Cumming et al., 1983*), and thus would confound the results.

While it has been shown that some histone modifications such as H3K4me3 can be transmitted transgenerationally through sperm (*Lismer et al., 2020*; *Siklenka et al., 2015*), there is also a growing body of evidence demonstrating that physiological stress leads to changes in the signature of non-coding RNA within the sperm (*Gapp et al., 2014*; *Chan et al., 2020*; *Gapp et al., 2021*; *Gapp et al., 2020*; *Rodgers et al., 2013*; *Rodgers et al., 2015*). In one study, small RNAs were shown to be delivered to maturing sperm from extracellular vesicles originating in the somatic cells of the epididymis, which are the likely sensors of glucocorticoids (*Chan et al., 2020*). Our finding that perinatal dex treatment increases GR activity in developing prospermatogonia and leads to altered RNA splicing could confer an evolutionary advantage by increasing the overall transcript diversity of the germline in times of stress, allowing for increased adaptation of the germline to selective pressures from the environment. Distinct from the effects of stress on small RNAs originating from epididymal epithelial cells (*Chan et al., 2020*), this raises the question whether direct activation of GR signaling in male germ cells could lead to alterations in small non-coding RNAs, and whether such changes could be heritable across generations.

## Conditional deletion of GR using Prdm1-Cre does not affect male fertility

As proper RNA splicing is crucial for meiotic progression in the spermatogenic lineage (*Kuroda et al., 2000*; *Li et al., 2007*; *O'Bryan et al., 2013*; *Schmid et al., 2013*; *Zagore et al., 2015*; *Naro et al., 2017*; *Liu et al., 2017*; *Xu et al., 2017*; *Horiuchi et al., 2018*; *Ehrmann et al., 2019*; *Legrand et al., 2019*; *Yuan et al., 2021*; *Wu et al., 2021*), perturbations to RNA splicing as a result of elevated GR signaling could have important consequences for fertility. To our surprise, genetic deletion of GR specifically in germ cells of the testis did not measurably impact fertility (*Figure 6B*). While the subfertility caused by the presence of the *Prdm1*-Cre transgene alone could be masking a more subtle subfertility phenotype caused by loss of GR, it is also possible that homeostatic levels of GR signaling do not regulate splicing. While pharmacologic agonism of GR signaling in the adult testis could lead to more pronounced disruptions to splicing and subsequent fertility compared to deletion of GR, we were unfortunately unable to test this possibility. Both chronic stress and dex administration have been shown to inhibit testosterone production by Leydig cells (*Bambino and Hsueh, 1981*; *Hales and Payne, 1989*; *Kavitha et al., 2006*; *Orr and Mann, 1992*; *Orr and Mann, 1990*; *Rengarajan and Balasubramanian, 2007*; *Rengarajan and Balasubramanian, 2008*; *Saez et al., 1977*; *Sankar et al., 2000*; *Marić et al., 1996*), and given that spermatogenesis requires testosterone (*Smith and Walker,*

*2014*), systemic dex administration would not tease apart the cell intrinsic effects of GR activation in spermatogonia. Future studies using transgenic overexpression of GR in the spermatogonia could address this effect of elevated GR signaling on fertility.

## Materials and methods

### Mouse husbandry

All animal work were performed under strict adherence to the guidelines and protocols set forth by the University of California San Francisco's Institutional Animal Care and Use Committee (IACUC), and all experiments were performed in an animal facility approved by the Association for the Assessment and Accreditation of Laboratory Animal Care International (AAALAC). All procedures performed were approved by the UCSF IACUC under protocol numbers AN169770 and AN200504. All mice were maintained in a temperature-controlled animal facility with 12 hr light dark cycles, and were given access to food and water ad libitum.

### Mouse timed pregnancies

All matings were set in the evenings (15:00 or after), and the presence of a vaginal plug the morning after mating (08:00 – 11:00) was denoted embryonic day 0.5 (E0.5). Pregnant females were dissected at various timepoints, the uterine horns removed into ice-cold 0.4% BSA in PBS, and embryos dissected and staged based on canonical morphologic features. For all postnatal timepoints, postnatal day 0 (PN0) was assigned as the morning a litter was first seen (where litters were dropped the night of E18.5).

### Mouse fertility test

Adult male mice (age 6 weeks) were paired with WT, mixed background females (age 6 weeks), and females were monitored daily for the presence of a vaginal plug to indicate mating. For each pairing, number of litters produced and total number of pups per litter were recorded over an 8-week period.

### Genotyping

Total genomic DNA was extracted from ear punches or tail tips by boiling in alkaline lysis buffer (25 mM NaOH; 0.2 mM EDTA) for 45 minutes at 95 °C, cooling to 4 °C, and then neutralizing with an equal volume of 40 mM Tris-HCl. All genotyping reactions were performed with KAPA HotStart Mouse Genotyping Kit (Roche, 07961316001) with 1 µL of gDNA and a final primer concentration of 5 µM each. Cycler conditions are listed below each table. All PCR products were separated by electrophoresis on a 2% agarose gel in TBS stained with ethidium bromide (VWR, E3050), and genotypes determined based on sizes of DNA products.

### Mouse lines used, genotyping primers, and genotyping reactions

| Allele | Primer Sequences | Expected Sizes | MGI# | Rxn |
|---|---|---|---|---|
| *Nr3c1*tm1.1Jda | 5'-CAGGTATTGGTGCTTGTTAGCACTT-3' | WT: 155 bp<br>KO: 208 bp<br>Flox: 318 bp | *5447468* | *A* |
| | 5'-GCCTGCATCTTTTACATGTGTTGTTTCC-3' | | | |
| | 5'-CAGCTTACAGGATAGCCAGTGATATCTGT-3' | | | |
| *Pou5f1*tm1.1(cre/Esr1*)Yseg | 5'-GCTTTCTCCAACCGCAGGCTCTC-3' | WT: 234 bp<br>Cre: 169 bp | *5049897* | *A* |
| | 5'-CCAAGGCAAGGGAGGTAGACAAG-3' | | | |
| | 5'-GCCCTCACATTGCCAAAAGACGG-3' | | | |

*Continued on next page*

*Continued*

| Allele | Primer Sequences | Expected Sizes | MGI# | Rxn |
|---|---|---|---|---|
| Tmem163$^{Tg(ACTB-cre)2Mrt}$ | 5'-TGCAATCCCTTGACACAGA-3' | WT: 241 bp<br>Cre: 187 bp | *2176050* | *A* |
| | 5'-ACCAGTTTCCAGTCCTTCTGG-3' | | | |
| | 5'-GTCCTTACCCAGAGTGCAGGT-3' | | | |
| Tg(Pou5f1-EGFP)2Mnn | 5'-GCACGACTTCTTCAAGTCCGCCATGC-3' | Tg: 270 bp | *3057158* | *B* |
| | 5'-GCGGATCTTGAAGTTCACCTTGATGCC-3' | | | |
| Tg(Prdm1-cre)1Masu | 5'-GCCGAGGTGCGCGTCAGTAC-3' | Tg: 200 bp<br>Ctrl: 324 bp | *3586890* | *C* |
| | 5'-CTGAACATGTCCATCAGGTTCTTG-3' | | | |
| | 5'-CTAGGCCACAGAATTGAAAGATCT-3' | | | |
| | 5'-GTAGGTGGAAATTCTAGCATCATCC-3' | | | |

Cycler conditions:
1. 94 °C, 30 sec – 1 cycle // (94 °C, 15 sec; 60 °C, 15 sec; 72 °C, 30 sec) – 35 cycles // (72 °C, 10 min) – 1 cycle // 4 °C hold.
2. 93 °C, 1 min – 1 cycle // (93 °C, 20 sec; 68 °C, 3 min) – 30 cycles // (72 °C, 10 min) – 1 cycle // 4 °C hold.
3. 94 °C, 30 sec – 1 cycle // (94 °C, 15 sec; 66 °C, 15 sec; 72 °C, 30 sec) – 35 cycles // (72 °C, 10 min) – 1 cycle // 4 °C hold.

## Dex in vivo dosing

For all in vivo dex dosing experiments, a water-soluble form of dex (Sigma, D2915) was utilized to allow for delivery in a saline vehicle (0.9% NaCl w/v in ddH$_2$O). The developmental time windows of dex administration were determined based on GR's germline expression profile, and differed for the male and female germline. Dex dose and time widows varied between experiments, and are outlined for each experiment in the Results section. For all injections, the volume given to both saline and dex-treated animals was fixed, where the concentration of dex in the injected solution was scaled to ensure precise dose delivery (normalized to body weight) in the fixed volume. For gestational timepoints (E12.5 - E18.5), pregnant dams were injected via IP injection at approximately 09:00 with 100 µL saline/dex. For postnatal timepoints (PN0 - PN21), pups were weighed and injected subcutaneously (in the back flank) with 20 µL of saline or dex. See experimental schematics for exact doses and timing.

## Ex vivo gonad culture

Fetal ovaries were dissected at E14.5 and pooled into ice cold 0.4% BSA in PBS as described above. Ovaries were cultured in Millicell 24-well hanging inserts, 1.0 µm PET (Millipore, MCRP24H48) at 37 °C, 5% CO$_2$. Ovaries were cultured in DMEM / F12 base medium (Gibco, 11330–032) supplemented with 10% charcoal-stripped FBS (Sigma, F6765), 1 mM sodium pyruvate (Gibco, 11360070), 0.5 X MEM Nonessential Amino Acids (Corning, 25–025 CI), and 100 U/mL pen/strep (Gibco, 15140–122). Bottom chambers were filled with 1.3 mL of media, top inserts with 200 µL of media, and the media was changed daily. For experimental conditions, dex (Sigma, D1756) dissolved in DMSO was added to cultures at a final concentration of 1.0 µM.

## Embryonic gonad digestion

Fetal gonads were dissected into ice cold 0.4% BSA in PBS, washed once, and maintained on ice until ready for digestion. 0.4% BSA solution was removed and replaced with 100 µL of 0.25% Trypsin-EDTA (Fisher Sci, 25200056) per ovary pair, or 150 µL per testis pair. Samples were incubated in a 37 °C water bath for 30 min, with gentle pipetting every 10–15 min to facilitate the dissociation. After 30 min, DNase I (1 mg / mL) was added at a 1:10 dilution, and samples were incubated another 10 min at 37 °C. Samples were pipetted to ensure complete digestion, and then an equal volume of ice-cold FBS (Gibco, 10437028) was added to inactivate trypsin. To prepare for FACS sorting, Sytox Blue viability dye (Invitrogen, S34857) was added to samples at 1:1000 dilution, and then samples were filtered through a 35 µm filter into FACS tubes (Falcon, 352235).

## Fluorescence activated cell sorting (FACS)

To isolate germ cells expressing the *Pou5f1-ΔPE-eGFP* transgene from the GFP⁻ somatic compartment, we utilized either a BD FACSAria II or III system. Briefly, cells were gated to remove doublets/multiplets via FSC height vs area comparison, and then dead cells were gated out based on uptake of the Sytox Blue viability dye. GFP⁺ germ cells and GFP⁻ somatic cells were sorted directly into 300 μL of QIAgen RLT Plus Buffer (with βME). Upon completion of the sort, samples were immediately vortexed for 30 s and flash frozen.

## Tissue fixation

All mouse gonad tissue was fixed with fresh 4% paraformaldehyde (PFA) in PBS at 4 °C with rocking. Fixation times were as follows:

### Tissue fixation parameters

| Sex | Stage | Vol 4% PFA | Fixation Time | PBS Washes |
|-----|-------|-----------|---------------|------------|
| XY | E12.5 - E14.5 | 1 mL / litter | 30 min | 3x10 min |
| | E15.5 - E16.5 | 1 mL / litter | 45 min | 3x10 min |
| | E17.5 - E18.5 | 1 mL / litter | 1 hr | 3x10 min |
| | PN0 - PN2 | 1 mL / pair | 2 hr | 4x10 min |
| | PN3 - PN5 | 1 mL / pair | 4 hr | 4x10 min |
| | PN6 - PN10 | 1 mL / testis | 6 hr | 4x15 min |
| | PN11 - PN14 | 4 mL / testis | 12 hr | 4x15 min |
| | PN15 - PN17 | 4 mL / testis | 15 hr | 4x15 min |
| | PN18 - PN21 | 4 mL / testis | 18 hr | 4x15 min |
| | Adult | 4 mL / testis | O/N | 4x15 min |
| XX | E12.5 - E15.5 | 1 mL / litter | 30 min | 3x10 min |
| | E16.5 - E18.5 | 1 mL / litter | 45 min | 3x10 min |
| | PN0 - PN7 | 1 mL / pair | 2 hr | 4x10 min |
| | PN21 - Adult | 4 mL / ovary | O/N | 4x15 min |

## Tissue embedding and sectioning

Following PFA fixation and PBS washes, embryonic gonads were incubated overnight in 30% sucrose (in PBS) at 4 °C. The following day, the tissue was embedded in OCT (Tissue-Tek. 4583) and blocks stored at –80 °C until ready for sectioning. For all postnatal and adult tissues, gonads were first incubated in 10% sucrose (in PBS) for about 1–2 hr at 4 °C until the tissue sank to the bottom of the tube. Gonads were then transferred to incubate overnight in 30% sucrose (in PBS) at 4 °C. The following day, the tissue was transferred to 50/50 OCT / 30% sucrose and allowed to equilibrate for 6 hr at 4 °C prior to embedding in 100% OCT. Blocks were similarly stored at –80 °C until ready for sectioning. All blocks were sectioned on a Leica 3050 S Cryostat at a thickness of 5–10 μm depending on the tissue.

## Section immunofluorescence

For all cryosectioned slides, slides were thawed to room temperature and then washed three times (5 min each) with 1 X PBS to remove residual OCT. Slides were blocked for 1 hr at room temperature in 10% heat inactivated donkey serum +0.1% Triton X-100 in PBS. All primary antibody incubations were performed at 4 °C overnight in a humidified chamber, with primary antibodies diluted accordingly in blocking buffer. The next day, slides were washed three times with 1 X PBS (5 min each), followed by a 1 hr incubation at RT with secondary antibodies diluted in blocking buffer. Samples were washed three times with 1 X PBS (5 min each), mounted in VECTASHIELD Antifade Mounting Medium (Vector Laboratories, H-1000), and sealed with a coverslip.

## Preparation of meiotic spreads from embryonic ovaries

After dissection, ovary pairs were transferred to 1 mL centrifuge tubes containing 100 µL of dissociation buffer (0.025% Trypsin, 2.5 mg/mL collagenase, and 0.1 mg/mL DNase). Samples were incubated in a 37 °C water bath for 30 min, and were pipetted vigorously every 10 min using a P-200 pipette to facilitate digestion. After 30 min, trypsin activity was stopped by adding an equal volume of FBS and mixing well. Next, and equal volume of freshly prepared hypotonic buffer (30 mM Tris pH 8.2, 50 mM sucrose, 17 mM sodium citrate, 5 mM EDTA, 0.5 mM DTT, 0.5 mM PMSF, at a final pH of 8.2) was added per sample, and samples were incubated for 30 min at room temperature. Samples were centrifuged for 10 min at 1000 rpm at room temperature, the supernatant removed, and then resuspended in 100 mM sucrose (+7.5 mM Boric acid, final pH 8.2). Samples were typically resuspended in 60 µL per ovary, which usually made three slides per ovary. Positively charged slides were pre-cleaned with 70% ethanol, and then dried gently with a Kimwipe. Using a hydrophobic pen, a small (~2cm x 2cm) square was drawn on each slide, and the squares were each coated with approximately 20 µL of fixative solution (1% PFA, 0.15% Triton X-100, 3 mM DTT, 7.5 mM Boric acid, final pH 9.2). Using a P-20 pipette, 20 µL of the cell suspension was dropped onto the square from approximately one foot above. Slides were incubated for 1 hr in a covered humidified chamber at room temperature to allow fixation of nuclei to slide, and then incubated uncovered for approximately 3 hr to allow slides to fully dry. Once dry, slides were washed twice with 0.4% Photoflo in $H_2O$ (Kodak, 146 4510) by fully submerging slides for 2 min each. Finally, slides were allowed to fully dry, then stored at –80 °C until ready to stain.

## Staining of meiotic spreads – SYCP1, SYCP3, and γH2AX co-stain

Slides were warmed to room temperature, and then immediately incubated with 0.1% Triton X-100 in PBS for 10 min to permeabilize. Slides were washed three times with 1 X PBS for 5 min each, and then blocked for 1 hr at room temp in 5% BSA in PBS. Slides were incubated overnight at 4 °C with primary antibodies diluted in 5% BSA: rabbit α-SYCP1 (Abcam, ab15090; 1:200) and mouse α-γH2AX (Millipore Sigma, 05–636-I; 1:200). The following day, slides were washed three times, ten minutes each, with 1 X PBS, and then incubated for 1 hr at room temp with secondaries in 5% BSA: donkey α-mouse 555 (1:200), donkey α-rabbit 647 (1:200), and DAPI (1:1000). Next, slides were washed three times, ten minutes each, with 1 X PBS, and then incubated for 1 hr at room temp with pre-conjugated mouse α-SYCP3 (Abcam, ab205846; 1:50). Slides were washed again three times, and then mounted with VECTASHIELD Antifade Mounting Medium (Vector Laboratories, H-1000).

## Confocal microscopy

All imaging was performed on a white-light Leica TCS SP8 inverted confocal microscope using either an HC FLUOTAR L 25×/0.95 VISIR water objective (Leica) or an HC PL APO 63 x/1.40 CS2 oil objective (Leica). All images were taken at 1024x1,024 pixel resolution, and any tile scan images were merged using Leica software.

## Quantitative image analysis using Imaris

All analyses described below were performed using Imaris Microscopy Image Analysis Software v8.3.1 (Oxford Instruments).

## Cellular segmentation with WGA

In order to segment individual cells within a tissue section, we utilized fluorophore-conjugated forms of the lectin wheat germ agglutinin (WGA) (Biotium, 29022–1, 29062–1, 29025–1) to outline all cellular membranes. Slides were stained overnight with WGA (1:50) using the standard section immunofluorescence protocol outlined above. Confocal.lif files were imported into Imaris, and individual cells were computationally segmented using the 'Cell' module with the following settings: { Detection Type: 'Detect Cell Only' // Region: 'Whole Image' // Cell Type: 'Cell Membrane' // Cell Smallest Diameter: '7 µm' // Cell Membrane Detail: '0.75 µm' // Cell Filter Type: 'Smooth' // Quality Threshold: 'Manual' // Number of voxels:>20.0 }. The resulting segmented regions were then filtered to remove 'empty' (non-cell-containing) regions by applying the following filters: { Size Filter: '25–400 µm²' // Cell Intensity Mean (DAPI channel):>20.0 }. Lastly, regions containing doublet cells (based on DAPI),

were manually removed. To further subset out germ cells, an additional filter was applied based on the germ cell marker used: { Cell Intensity Mean (Germ cell marker channel): *manually determined* }.

## Calculating normalized GR protein levels

After performing cellular segmentation with WGA as described above, the mean fluorescence intensity values of all channels were exported for each individual cell. Mean fluorescence intensity values for the GR channel were normalized to those of the DAPI channel on a per cell basis, and the resulting normalized values were plotted.

## Protein extraction and quantification

Proteins were extracted from either whole-gonad tissue or FACS-sorted cells using Pierce RIPA Buffer (Thermo Fisher, 89900) as per the manufacturer's protocol. For FACS-sorted, frozen cell pellets (100–300 k cells), pellets were thawed on ice and resuspended in 100 µL of RIPA buffer. Suspension was incubated on ice for 15 min, with vortexing every three minutes. Suspension was spun at 16,000 x $g$ for 15 min at 4 °C to pellet non-soluble cellular debris, and then supernatant transferred to a fresh tube. For whole-gonad tissue, dissected gonads were washed once with 1 X PBS (without BSA), and then resuspended in 100–200 µL of RIPA buffer (depending on stage / number of gonads). Tissue was homogenized using a motorized pestle, on ice, until no visible tissue pieces remained. The suspension was then incubated on ice for 15 min (with vortexing every 3 min), and spun at 16,000 x $g$ for 15 min at 4 °C as described above. Total amount of protein was quantified using a Pierce BCA Protein Assay Kit (Thermo Fisher, 23227) with BSA protein standards, following the manufacturer protocol exactly.

## Western blotting

Prior to loading, protein samples were diluted with sample buffer (4 X Laemmli Buffer (Bio-Rad, #1610747) supplemented with 10% β-mercaptoethanol) and water to reach a final sample buffer concentration of 1 X. Samples were boiled in microcentrifuge tubes in a thermocycler for 5 min at 95 °C to denature proteins. For all blots, samples were run on Tris/Glycine Mini-PROTEAN TGX Precast SDS-PAGE gels, 4–15% (Bio-Rad #4561085), with 1 X Tris/Glycine/SDS Buffer (Bio-Rad #1610732). Five µL of either WesternSure Pre-stained Chemiluminescent Protein Ladder (LI-COR #926–98000) or Chameleon Duo Pre-stained Protein Ladder (LI-COR #928–60000) were used depending on the downstream imaging system. All gels were electrophoresed at 50 V for 10 min, and then 100 V for 50–60 min. All gels were transferred to Immobilon-FL PVDF Membrane (Millipore #IPFL10100; pre-activated with 100% methanol) using the Bio-Rad Trans-Blot SD semi-dry transfer cell (Bio-Rad # 1703940) at 15 V for 60 min at 4 °C. Transfer buffer was prepared as a 1 X final solution by mixing 7:2:1 of ddH$_2$O: methanol: 10 X Tris/Glycine Transfer Buffer (Bio-Rad #1610734). Following transfer, membranes were blocked for 60 min with 5% non-fat milk (Bio-Rad #1706404) prepared in 1 X TBST (TBS +0.1% Tween 20; Bioworld #42020084–1) on a rotator at RT. Primary antibodies were diluted in 5% non-fat milk and membranes incubated at 4 °C overnight with gentle rotation. The following day, membranes were washed three times with 1 X TBST for 5 min at RT with gentle rotation. Secondary antibodies, HRP-conjugated goat α-rabbit IgG (Abcam #ab205718) or HRP-conjugated goat α-mouse IgG (Abcam #ab205719) were diluted 1:5000 in 5% non-fat milk and incubated with membranes for 1 hr at RT with gentle rotation, and then washed three times with 1 X TBST for 5 min at RT with gentle rotation. Membranes were imaged using the Pierce ECL Western Blotting Substrate (Thermo #32109) and developed on autoradiography film.

## RNA isolation

For bulk gonad tissue (both freshly isolated and ex vivo cultured), gonads were first homogenized in a 1.5 mL centrifuge tube in 100 µL TRIzol (Invitrogen 15596026) using a motorized pestle. After bringing the final volume to 1 mL with TRIzol, samples were vortexed on high for 30 s before storing at –80 °C. To extract total RNA from TRIzol samples, 200 µL of chloroform was added per sample. Samples were vortexed vigorously for 30 s and then centrifuged at 12,000 x $g$ for 15 min at 4 °C. The upper aqueous layer was carefully extracted, mixed 1:1 with an equal volume of fresh 70% ethanol, and loaded onto a QIAgen RNeasy Micro spin column for further cleanup and elution. The QIAgen RNeasy Micro Kit (cat. no. 74004) protocol was followed exactly, including the on-column DNase digestion for 15 min. For FACS-sorted cell samples that were sorted into QIAgen RLT Plus Buffer, samples were mixed with

equal volume of 70% ethanol and loaded onto column without TRIzol homogenization and chloroform extraction. RNA concentrations were quantified using the Qubit RNA HS Assay Kit (ThermoFisher cat. no. Q32852) following the manufacturer's protocol.

## SYBR green qRT-PCR primer design

All primer sets used for qRT-PCR were designed from scratch. For each gene of interest, the mouse cDNA sequence of the predominant, full-length isoform was downloaded from the Ensembl database (https://www.ensembl.org). To identify potential primers with minimal off-target amplification across the mouse transcriptome, cDNA sequences were uploaded to NCBI's Primer BLAST tool (https://www.ncbi.nlm.nih.gov/tools/primer-blast), and primer searches were run with the following modified search parameters: PCR product size: 70–200 bp; Primer melting temperatures: Min: 60 °C, Opt: 62 °C, Max: 64 °C, Max $T_m$ diff: 1 °C; Database: Refseq mRNA; Organism: *Mus musculus*; Primer specificity stringency: "Ignore targets that have more than 4 or more mismatches to the primer"; Primer GC content: 30–70%; Max self complementarity: 3.00; Max pair complementarity: 3.00; Concentration of divalent cations: 3; Concentration of dNTPs: 0.2. Resulting putative primer pairs were screen manually for minimal self/pair complementarity scores, and then checked using the Beacon Designer tool (http://www.premierbiosoft.com/qOligo/Oligo.jsp?PID=1) for any potential primer-dimer events by selecting for primer pairs that had ΔG values closest to zero for Cross Dimer, Self Dimer and Hairpin scores. Lastly, the expected PCR product of any putative primer pairs was checked for potential secondary structure formation using IDT's UNAFold program (https://www.idtdna.com/unafold). As no primer pairs were completely without potential secondary structure formation, primer pairs were deemed acceptable so long as none of the predicted secondary structures had an anticipated $T_m$ value at or above the annealing temperature of our cycler reaction (60 °C). Whenever possible, primer pairs that spanned an intron were selected to minimize any amplification from contaminating genomic DNA. All primers used were validated through a standard melt curve analysis following qRT-PCR reactions (95 °C, 30 sec // 60 °C, 1 min // 95 °C then dec by 0.15 °C every sec) to ensure the amplification of a single PCR product as anticipated, and any primer pairs that failed melt curve analysis were discarded and redesigned.

## Quantitative RT-PCR

Prior to qRT-PCR, 100–1000 ng of RNA per sample was reverse transcribed using the qScript cDNA SuperMix (QuantaBio cat. no. 95048–100). qRT-PCR was performed using PowerUp SYBR Green Master Mix (Thermo A25776), as per manufacturer's instructions. For each individual reaction, 1 ng total cDNA was used, with a final primer concentration of 0.5 µM each, and all reactions were performed in technical triplicate. All experiments were run on a Thermo Fisher QuantStudio 5 with the following cycler conditions: (50 °C, 2 min) – 1 cycle // (95 °C, 2 min) – 1 cycle // (95 °C, 30 sec; 60 °C, 1 min) – 40 cycles. For all experiments, qRT-PCR analysis was performed using standard $\Delta\Delta C_T$ method. Briefly, $\Delta C_T$ values were calculated by normalizing $C_T$ values for individual target probes to the average $C_T$ value of a standard housekeeping gene (typically 18 S ribosomal RNA); $\Delta\Delta C_T$ values were calculated by normalizing sample $\Delta C_T$ values to the reference sample of choice (here usually 'WT' or dex untreated); Fold change values were calculated by taking $2^{-\Delta\Delta CT}$.

## Qualitative RT-PCR

Prior to RT-PCR, 100–1000 ng of RNA per sample was reverse transcribed using the qScript cDNA SuperMix (QuantaBio, 95048–100). All PCR reactions were performed using GoTaq Green Master Mix (Promega, M7123) with 0.25 ng total cDNA per reaction, with a final primer concentration of 0.5 µM each. Cycler conditions: { 95 °C, 3 min – 1 cycle // (95 °C, 15 sec; (55-60)°C, 15 sec; 72 °C, 30 sec) – 35 cycles // (72 °C, 10 min) – 1 cycle // 4 °C hold }. For each primer pair, a temperature gradient PCR was used to determine the optimal annealing temperature. Products were electrophoresed on a 2% agarose gel at 150 V for 30 min, and then visualized using a Bio-Rad GelDoc imager. Mouse XpressRef Universal Total RNA (QIAgen, 338114) was reverse transcribed and used at a concentration of 250 ng cDNA per reaction as a positive control for each primer set. Nuclease free water (Thermo Fisher, AM9937) was used as a negative control.

## Single-cell RNA sequencing
### Sample preparation

To generate embryos homozygous for the GR knockout allele (*Nr3c1*^KO/KO), heterozygous females (*Nr3c1*^KO/+) were crossed to heterozygous males also carrying an *Pou5f1*-GFP transgene (*Nr3c1*^KO/+;

Tg:*Pou5f1*$^{GFP/GFP}$) to facilitate FACS sorting of germ cells. Pregnant dams were dissected the morning of E15.5, and tail clips were taken from each embryo to determine GR genotype. Fetal ovaries were dissected on ice, the mesonephroi removed, and the ovaries were digested for FACS sorting as described above. Fetal ovaries were pooled based on genotype (n=4 ovary pairs for WT GR$^{+/+}$, and n=2 ovary pairs for *Nr3c1*$^{KO/KO}$). In order to enrich for germ cells (relative to the predominant somatic cells), live germ cells were FACS sorted based on Tg:*Pou5f1*$^{GFP}$ expression, and somatic cells spiked back into the germ cell population at a ratio of 60 : 40, germ: soma. The final cell suspension was resuspended in 0.04% BSA at a concentration of 1000 cells / µL, and was used for 10 X single-cell capture as described below.

## 10X capture, library preparation, and sequencing

The following sample preparation and sequencing was all performed by the UCSF CoLabs Initiative. All samples were processed with the standard Chromium 10 X Single-Cell 3' Reagent Kit v3 workflow. In summary, final cell suspensions were loaded onto the 10 X Chromium microfluidics chip along with reaction master mix, partitioning oil, and Single-Cell 3' v3 Gel Beads in order to generate Gel Beads-in-emulsion (GEMs) containing individual cells. RT reactions were performed within individual GEMs to generate cell-barcoded full-length cDNA, followed by GEM breakdown, cDNA pooling, cDNA PCR amplification, cleanup, and QC. Amplified cDNA was enzymatically fragmented, end repaired, A-tailed, size selected, and ligated with adapters. Following a sample indexing PCR reaction and final size selection, sample libraries were pooled, QC'ed, and sequenced on an Illumina NovoSeq 6000 platform. Resulting paired-end sequencing reads were processed using the 10 X Cell Ranger software to generate feature-barcode matrices.

## Data analysis

All single-cell RNA-seq data processing was performed in house using Seurat (*Satija et al., 2015*) v3.2.3. Count matrices for both WT and KO conditions were read into Seurat to create Seurat Objects, and were then filtered as follows: WT: { nFeature_RNA: 1500–6500 // nCount_RNA: 5500–40,000 // % mitochondria: < 10% }; KO: { nFeature_RNA: 1200–5500 // nCount_RNA: 3000–30,000 // % mitochondria: < 10% }. Both WT and KO Seurat objects were combined using the 'merge' function, and the resulting objects were processed for dimensional reduction using the standard Seurat workflow using the following functions: NormalizeData, FindVariableFeatures, ScaleData, RunPCA, RunUMAP, FindNeighbors, and FindClusters. During ScaleData, the following variables were regressed out: nFeature_RNA, nCount_RNA, percent.mito, and percent.ribo. The cell type identities of each cluster were determined by comparing top cluster markers to a previously annotated list E16.5 ovarian cell type specific markers (*Niu and Spradling, 2020*). Pairwise differential gene expression analyses were performed between WT and KO cells independently for each of the main cell type clusters (germ, theca, granulosa, and epithelia) using the FindMarkers function using the default Wilcoxon Rank Sum test. Genes with an adjusted p-value less than 0.05 and a log fold change greater than 0.25 were considered significant.

## 3' Tag-Seq

### Sample preparation

To generate embryos homozygous for the GR knockout allele (*Nr3c1*$^{KO/KO}$), heterozygous females (*Nr3c1*$^{KO/+}$) were crossed to heterozygous males also carrying an *Pou5f1*-GFP transgene (*Nr3c1*$^{KO/+}$; Tg:*Pou5f1*$^{GFP/GFP}$) to facilitate FACS sorting of germ cells. Pregnant dams were dissected the morning of E17.5, and tail clips were taken from each embryo to determine GR genotype. Fetal ovaries were dissected on ice, the mesonephroi removed, and the ovaries were digested for FACS sorting as described above. Tg:*Pou5f1*-GFP$^+$ germ cells from individual embryo ovary pairs were FACS sorted directly in QIAgen RLT +buffer and stored at –80 until ready for RNA extractions as outlined above. To generate embryos with a conditional deletion of GR (*Nr3c1*$^{cKO/cKO}$), females homozygous for an exon 3 floxed GR allele (*Nr3c1*$^{flox/flox}$) were crossed to similar males that were also heterozygous for the germ cell specific *Pou5f1*$^{CreERT2}$ allele (*Nr3c1*$^{flox/flox}$; *Pou5f1*$^{CreERT2/+}$; Tg:*Pou5f1*$^{GFP/GFP}$). Pregnant dams were injected at E10.5 with 125 µg / g tamoxifen to induce recombination and deletion of GR exon 3 specifically in the germ cells. Both *Nr3c1*$^{cKO/cKO}$ and control *Nr3c1*$^{flox/flox}$ embryos were collected from tamoxifen-injected dams, and thus were equally exposed to tamoxifen in utero. Pregnant dams were

dissected the morning of E17.5 and processed for FACS sorting as described for *Nr3c1*[KO/KO] females above.

## Library preparation and sequencing

All library preparations and sequencing were performed by the University of California, Davis DNA Technologies & Expression Analysis Core. Gene expression profiling was carried out using a 3' Tag-RNA-Seq protocol. Barcoded sequencing libraries were prepared using the QuantSeq 3' mRNA-Seq Library Prep FWD kit (Lexogen) for multiplexed sequencing according to the manufacturer recommendations. The library fragment size distribution was determined using microcapillary gel electrophoresis on a Bioanalyzer 2100 (Agilent), and libraries were quantified using a Qubit fluorometer (LifeTechnologies). Final libraries were pooled in equimolar ratios and sequenced on an Illumina HiSeq 4000.

## Data analysis

Data analysis of 3' Tag-Seq data was performed in house. Illumina universal adapters (AGATCGGA AGAG) were trimmed from fastq files using cutadapt (*Martin, 2011*). Paired-end reads were aligned to the mm10 genome using STAR (*Dobin et al., 2013*) v2.6.0, and counts files were generated using featureCounts (*Liao et al., 2014*) v1.6.3. Differential expression analysis was performed using edgeR (*Robinson et al., 2010*) v3.28.1 and limma (*Ritchie et al., 2015*) v3.42.2. The resulting p-values were adjusted using the Benjamini and Hochberg's approach for controlling the false discovery rate. Genes with an adjusted p-value ≤0.05 were assigned as differentially expressed.

## Bulk RNA-seq on in vivo dex-dosed gonads

### Sample preparation

For analysis of female germ cells by RNA-seq, pregnant dams were injected by IP at approximately 09:00 with either saline or 10 mg dex / kg weight daily at E12.5, E13.5, E14.5 and E15.5. Embryos were dissected at E15.5 at approximately 15:00, and fetal ovaries placed in ice-cold 0.4% BSA in PBS. Mesonephroi were carefully microdissected away from the ovary, and ovaries were then digested for FACS sorting as described above. For analysis of male germ cells by RNA-seq, pregnant dams were injected via IP at approximately 09:00 with either saline of 10 mg dex / kg weight daily at E17.5 and E18.5. Following delivery the evening of E18.5, PN0 pups were injected the following morning subcutaneously with either saline or 10 mg dex / kg bodyweight. Pups were then sacrificed at approximately 11:00 on PN1, and testes placed in ice-cold 0.4% BSA in PBS. The tunica vaginalis was carefully microdissected away from each testis, and testes were then digested for FACS-sorting as described above.

### Library preparation and sequencing

All library preparations and sequencing were performed by the company Novogene. mRNA was purified from total RNA using poly-dT magnetic beads. Following mRNA fragmentation, first strand cDNA synthesis was carried out using random hexamer primers, followed by second strand cDNA synthesis. Fragment ends were repaired, A-tailed, and ligated with sequencing adapters, followed by size selection, PCR amplification, and purification. Final libraries were quantified using Qubit and run on an Agilent Bioanalyzer system to ensure proper size distribution. Quantified libraries were pooled and sequenced on an Illumina platform. Clustering of the index-coded samples was performed according to the manufacturer's instructions, and library preparations were subsequently sequenced to generate paired-end reads.

## Data analysis

Initial data QC, genome alignment, and differential expression analysis was all carried out by Novogene.

## Alignment

The mm10 reference genome was indexed using Hisat2 (*Pertea et al., 2016*) v2.0.5. Cleaned, paired-end reads were aligned to the mm10 genome using the splice-aware aligner Hisat2 v2.0.5.

## Expression quantification

Counts of reads mapping to each gene were determined using featureCounts (*Liao et al., 2014*) v1.5.0-p3. FPKM values were calculated for each gene based on the length of the gene, sequencing depth, and read counts mapping to the gene.

## Differential expression analysis

Differential expression analysis of pairwise conditions was performed using DESeq2 (v1.20.0) (*Love et al., 2014*). The resulting p-values were adjusted using the Benjamini and Hochberg's approach for controlling the false discovery rate. Genes with an adjusted *P*-value ≤0.05 were assigned as differentially expressed.

## Gene ontology enrichment analysis

GO Term enrichment analysis was performed using the GO Consortium's online tool at: http://gene-ontology.org/. A list of significant genes (adjusted p-value ≤0.05) was uploaded and used to look for enriched biological processes, utilizing a Fisher's Exact test followed by Bonferroni correction for multiple testing. Lists for upregulated and downregulated genes were run independently. The results were sorted based on highest fold enrichment scores, and the top ten processes for each category plotted.

## Differential splicing analysis using rMATS

Differential transcript splicing analysis was performed using rMATS (*Shen et al., 2012*) version 4.1.2, with genome-aligned BAM files generated from paired end RNA-seq reads used as input. Comparisons were made between saline treated and dex treated germ cells from PN1 testes (*Figure 5A*), using the rMATS settings: -t paired; `--readLength` 150. Results were visualized as sashimi plots, generated with the program rmats2sashimiplot.

## Antibodies and labeling reagents used

| Target | Company | Product # | Dilution | Host Species | Clonality |
|--------|---------|-----------|----------|--------------|-----------|
| GR (*Nr3c1*) * | Cell Signaling | 12041 S | 1:200 (IF) | Rabbit, IgG | mono |
| GR (*Nr3c1*) ** | Cell Signaling | 3660 S | 1:1000 (WB) | Rabbit, IgG | mono |
| GFP | Aves | GFP-1020 | 1:200 (IF) | Chicken, IgY | poly |
| TRA98 (*Gcna1*) | Abcam | ab82527 | 1:300 (IF) | Rat, IgG2a | mono |
| NOBOX | N/A*** | N/A | 1:400 (IF) | Goat | unknown |
| SYCP3-488 | Abcam | ab205846 | 1:50 (spreads) | Mouse, IgG1 | mono |
| γH2AX (Ser139) | Millipore Sigma | 05–636-I | 1:200 (spreads) | Mouse, 1gG1 κ | mono |
| SYCP1 | Abcam | ab15090 | 1:200 (spreads) | Rabbit, IgG | poly |
| FKBP5 | ThermoFisher | 711292 | 1:200 (IF) | Rabbit, IgG | poly |
| PLZF (*Zbtb16*) | R&D | AF2944 | 1:200 (IF) | Goat, IgG | poly |
| c-KIT (*CD117*) | R&D | AF1356 | 1:200 (IF) | Goat, IgG | poly |
| VASA/MVH (*Ddx4*) | Abcam | ab13840 | 1:500 (IF) | Rabbit, IgG | poly |
| Lectin PNA | Thermo Fisher | L21409 | 1:1000 (IF) | N/A | N/A |

## RT-PCR primers used

| Target | Forward Primer | Reverse Primer |
|--------|----------------|----------------|
| *Actb* | 5'-ACTCTGTGTGGATCGGTGGC-3' | 5'-AGCTCAGTAACAGTCCGCCTAGAA-3' |
| *Ddx4* | 5'-AGGCCCGTTCAGAAGAGGGG-3' | 5'-GCCTGAATCACTTGCTGCTGGT-3' |

*Continued on next page*

*Continued*

| Target | Forward Primer | Reverse Primer |
|--------|---------------|----------------|
| *Nr3c1* - exon 1 A | 5'-GCCCGGCCTTATCTGCTAGAAGTGG-3' | 5'-ACAAGTCCATCACGCTTCCCCTCC-3' |
| *Nr3c1* - exon 1B | 5'-TATCTGGCTGCGGTGGGAGCC-3' | 5'-ACAAGTCCATCACGCTTCCCCTCC-3' |
| *Nr3c1* - exon 1 C | 5'-GGAGCCAGGGAGAAGAGAAACTAAAG-3' | 5'-ACAAGTCCATCACGCTTCCCCTCC-3' |
| *Nr3c1* - exon 1D | 5'-GACCTGGCAGCACGCGAGT-3' | 5'-ACAAGTCCATCACGCTTCCCCTCC-3' |
| *Nr3c1* - exon 1E | 5'-TTCGCCGTGCAACTTCCTCCGAAT-3' | 5'-ACAAGTCCATCACGCTTCCCCTCC-3' |
| *Nr3c1* - exon 1 F | 5'-CACTGAGCCTGGAGCAGCAAATG-3' | 5'-ACAAGTCCATCACGCTTCCCCTCC-3' |
| *Nr3c1* - exon 1 G | 5'-GAGGGCAGGCTTCCGTGACAAC-3' | 5'-ACAAGTCCATCACGCTTCCCCTCC-3' |
| *Nr3c1* - exon 1α | 5'-CCTTGCAGTTGCCGACAGTCG-3' | 5'-ACAAGTCCATCACGCTTCCCCTCC-3' |
| *Nr3c1* - exon 1β | 5'-CATAACACCTTACTCCCCAACCCCC-3' | 5'-ACAAGTCCATCACGCTTCCCCTCC-3' |
| *Nr3c1* - exon 1γ | 5'-GAGCACCTCTGCCAAAATGGTGAC-3' | 5'-ACAAGTCCATCACGCTTCCCCTCC-3' |
| *Nr3c1* - exon 2–3 | 5'-CAGCTCCTCCACAGCAACGG-3' | 5'-TGCTGTCCTTCCACTGCTCTTT-3' |

## qRT-PCR primers used

| Target | Forward Primer | Reverse Primer |
|--------|---------------|----------------|
| *Dmc1* | 5'-CGGCTACTCAGGTGGAAAGA-3' | 5'-GTTGAAGCGGTCAGCAATGT-3' |
| *Fkbp5* | 5'-ATGCTTATGGCTCGGCTGG-3' | 5'-AGTATCCCTCGCCTTTCCGT-3' |
| *Gapdh* | 5'-ACTTTGGCATTGTGGAAGGG-3' | 5'-AGGGATGATGTTCTGGGCA-3' |
| *Hnrnpa2b1* | 5'-CTTTCTCATCTCGCTCGGCT-3' | 5'-GTTCCTTTTCTCTCTCCATCGC-3' |
| *Hnrnph3* | 5'-AGGATACGGGTCTGTTGGGA-3' | 5'-TCCACTCATTCCACCTTGCC-3' |
| *Nr3c1* | 5'-CAGCTCCTCCACAGCAACGG-3' | 5'-TGCTGTCCTTCCACTGCTCTTT-3' |
| *Prdm9* | 5'-GCAGAGATGGGAGAGTGGGA-3' | 5'-TGGGGTTTCATTGCTTGCCT-3' |
| *Prpf31* | 5'-ACCCTGTCTGGCTTCTCTTC-3' | 5'-CACCTTCCCTTCTGTGCTCT-3' |
| *Rn18S* | 5'-TGCATGTCTAAGTACGCACGGC-3' | 5'-AGCGAGCGACCAAAGGAACC-3' |
| *Smc1b* | 5'-GCCCACCTTACACTCCTTCT-3' | 5'-TCTGGCTTCCTTTCTGCTGG-3' |
| *Srsf7* | 5'-GAGGATTGGATGGGAAAGTGAT-3' | 5'-CGTGACCTGCTTCTTCTTCG-3' |
| *Sycp1* | 5'-GAGGGGAAGCTCACGGTTC-3' | 5'-CAGTGTGAAGGGCTTTTGCT-3' |
| *Sycp3* | 5'-ATCTGGGAAGCCACCTTTGG-3' | 5'-GAGCCTTTTCATCAGCAACATCT-3' |
| *Tra2b* | 5'-GCTCCTCGCAAAAGTGTGG-3' | 5'-GATTCCCGCTCGCCGT-3' |

## Acknowledgements

The authors thank thesis committee members Barbara Panning, Kaveh Ashrafi and Brian Feldman for critical feedback over the course of this project. Thank you to Ryan Samuel for consulting on scRNA-seq analysis, and thank you to Bikem Soygur for exceptional scientific guidance and technical assistance on this project. We also thank the UC Davis DNA Technologies & Expression analysis core for 3' Tag-Seq library prep and sequencing, the UCSF CoLabs Initiative for performing Illumina 10 X capture, library prep, and sequencing, the UCSF Parnassus Flow Cytometry Core for assistance with cell sorting, as well as A Rajkovic for generously providing Nobox antibody. Funding: SAC was supported by the Ruth L Kirschstein National Research Service Award Individual Predoctoral F31 Fellowship NIH 5F31HD101234, NR is supported by the NIH Endocrinology T32-DK007418 Fellowship, MHF is supported by NIH 1F31HD110208 and the Hillblom/BARI Graduate Student Fellowship Award, and DJL is supported by P30-ES030284, R01ES028212, and

R01GM122902, the W.M. Keck Foundation, and the UCSF Program for Breakthrough Biomedical Research.

## Additional information

### Funding

| Funder | Grant reference number | Author |
|---|---|---|
| Eunice Kennedy Shriver National Institute of Child Health and Human Development | 5F31HD101234 | Steven A Cincotta |
| National Institute of Diabetes and Digestive and Kidney Diseases | T32-DK007418 | Nainoa Richardson |
| Eunice Kennedy Shriver National Institute of Child Health and Human Development | 1F31HD110208 | Mariko H Foecke |
| Hillblom Foundation | Graduate Student Fellowship | Mariko H Foecke |
| Bakar Aging Research Institute | Graduate Student Fellowship | Mariko H Foecke |
| National Institute of Environmental Health Sciences | P30-ES030284 | Diana J Laird |
| National Institute of Environmental Health Sciences | R01ES028212 | Diana J Laird |
| National Institute of General Medical Sciences | R01GM122902 | Diana J Laird |
| WM Keck Foundation | | Diana J Laird |
| University of California, San Francisco | Program for Breakthrough Biomedical Research | Diana J Laird |

The funders had no role in study design, data collection and interpretation, or the decision to submit the work for publication.

### Author contributions

Steven A Cincotta, Conceptualization, Data curation, Formal analysis, Investigation, Methodology, Writing - original draft, Writing - review and editing; Nainoa Richardson, Formal analysis, Investigation, Writing - review and editing; Mariko H Foecke, Investigation, Writing - review and editing; Diana J Laird, Conceptualization, Supervision, Funding acquisition, Writing - original draft, Writing - review and editing

### Author ORCIDs

Steven A Cincotta  http://orcid.org/0000-0003-3802-3673
Mariko H Foecke  http://orcid.org/0000-0002-9205-940X
Diana J Laird  http://orcid.org/0000-0002-4930-0560

### Ethics

All animal work was performed under strict adherence to the guidelines and protocols set forth by the University of California San Francisco's Institutional Animal Care and Use Committee (IACUC) protocols AN169770 and AN200504, and all experiments were performed in an animal facility approved by the Association for the Assessment and Accreditation of Laboratory Animal Care International (AAALAC). All mice were maintained in a temperature-controlled animal facility with 12 hour light dark cycles, and were given access to food and water ad libitum.

Joint Public Review: https://doi.org/10.7554/eLife.90164.3.sa1
Author Response https://doi.org/10.7554/eLife.90164.3.sa2

## Additional files

### Supplementary files
• MDAR checklist

### Data availability
Genomics data has been deposited in the Gene Expression Omnibus (GEO) under the accession code GSE234681.

The following dataset was generated:

| Author(s) | Year | Dataset title | Dataset URL | Database and Identifier |
|---|---|---|---|---|
| Cincotta SA | 2023 | Differential susceptibility of male and female germ cells to glucocorticoid-mediated signaling | https://www.ncbi.nlm.nih.gov/geo/query/acc.cgi?acc=GSE234681 | NCBI Gene Expression Omnibus, GSE234681 |

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
