## [Editor Report · eLife assessment]

This work reports a **valuable** finding on glucocorticoid signaling in male and female germ cells in mice, pointing out sexual dimorphism in transcriptomic responsiveness. The **convincing** evidence provided supports an inert GR signaling despite the presence of GR in the female germline and GR-mediated alternative splicing in response to dexamethasone treatment in the male germline. The work may interest basic researchers and physician-scientists working on reproduction and stress-related disease conditions.

---

## [Referee Report · Joint Public Review]

Summary:

Cincotta et al set out to investigate the presence of glucocorticoid receptors in the male and female embryonic germline. They further investigate the impact of tissue specific genetically induced receptor absence and/or systemic receptor activation on fertility and RNA regulation. They are motivated by several lines of research that report inter and transgenerational effects of stress and or glucocorticoid receptor activation and suggest that their findings provide an explanatory mechanism to mechanistically back parental stress hormone exposure induced phenotypes in the offspring.

Strengths:

- A chronological immunofluorescent assessment of GR in fetal and early life oocyte and sperm development.

- RNA seq data that reveal novel cell type specific isoforms validated by q-RT PCR E15.5 in the oocyte.

- 2 alternative approaches to knock out GR to study transcriptional outcomes. Oocytes: systemic GR KO (E17.5) with low input 3-tag seq and germline specific GR KO (E15.5) on fetal oocyte expression via 10X single cell seq and 3-cap sequencing on sorted KO versus WT oocytes - both indicating little impact on polyadenylated RNAs -

- 2 alternative approaches to assess the effect of GR activation in vivo (systemic) and ex vivo (ovary culture): here the RNA seq did show again some changes in germ cells and many in the soma.

- They exclude oocyte specific GR signaling inhibition via beta isoforms

- Perinatal male germline shows differential splicing regulation in response to systemic Dex administration, results were backed up with q-PCR analysis of splicing factors.

Weaknesses:

- Sequencing techniques used are not Total RNA but either are focused on all polyA transcripts (10x) - effects on non-polyA-transcripts are left unexplored.

The number of replicates in the low input seq is very low and hence this might be underpowered. Since Dex treatment showed some (modest) changes in oocyte RNA effects of GR depletion might only become apparent upon Dex treatment as an interaction. Meaning GR activation in the presence of GR shows changes, upon GR depletion those changes are abolished  statistically speaking an interaction  conclusion: there are germline GR effects that get abolished when there is no GR hinting on germline GR autonomous effects.

- Effects in oocytes following systemic Dex might be indirect due to GR activation in the soma. The changes observed might be irrelevant to meiosis and thus in the manuscript are deemed irrelevant, but they could still lead to settle consequences. in other terms.

Even though ex vivo culture of ovaries shows GR translocation to nucleus it is not sure whether the in vivo systemic administration does the same. The authors argue in their rebuttal that GR is already nuclear in fetal oocytes hence the

conclusion that fetal oocytes are resistant to GR manipulation is understandable, at least for the readouts that were considered. Yet the question arises: If GR is already nuclear (active) in the absence of additional Dex treatment why does GR knock out not elicit any changes in the considered readouts -> what are we missing.

This work is a good reference point for researchers interested in glucocorticoid hormone signaling fertility and RNA splicing. It might spark further studies on germline-specific GR functions and the impact of GR activation on alternative splicing.

The study provides a characterization of GR and some aspects of GR perturbation, and the negative findings in this study do help to rule out a range of specific roles of GR in the germline. This will help the study of unexplored options. The authors do acknowledge the unexplored options in their discussion.

The intro of the study eludes to implications for intergenerational effects via epigenetic modifications in the germline and points out additional potential indirect effects of reproductive tissue GR signaling on the germline. Future studies might hence focus on further exploration of epigenetic modifications and/or indirect effects.

---

## [Author Response]

The following is the authors’ response to the original reviews.

We are very grateful to both reviewers for taking the time to review our manuscript and data in great detail. We thank you for the fair assessment of our work, the helpful feedback, and for recognizing the value of our work. We have done our best to address your concerns below:

eLife assessmentThis work reports a valuable finding on glucocorticoid signaling in male and female germ cells in mice, pointing out sexual dimorphism in transcriptomic responsiveness. While the evidence supporting the claims is generally solid, additional assessments would be required to fully confirm an inert GR signaling despite the presence of GR in the female germline andGR-mediated alternative splicing in response to dexamethasone treatment in the male germline. The work may interest basic researchers and physician-scientists working on reproduction and
**Public Reviews:**

**Reviewer #1 (Public Review):**
Summary:Cincotta et al set out to investigate the presence of glucocorticoid receptors in the male and female embryonic germline. They further investigate the impact of tissue-specific genetically induced receptor absence and/or systemic receptor activation on fertility and RNA regulation. They are motivated by several lines of research that report inter and transgenerational effects of stress and or glucocorticoid receptor activation and suggest that their findings provide an explanatory mechanism to mechanistically back parental stress hormone exposure-induced phenotypes in the offspring.Strengths:A chronological immunofluorescent assessment of GR in fetal and early life oocyte and sperm development.RNA seq data that reveal novel cell type specific isoforms validated by q-RT PCR E15.5 in the oocyte.2 alternative approaches to knock out GR to study transcriptional outcomes. Oocytes: systemic GR KO (E17.5) with low input 3-tag seq and germline-specific GR KO (E15.5) on fetal oocyte expression via 10X single cell seq and 3-cap sequencing on sorted KO versus WT oocytes both indicating little impact on polyadenylated RNAs2 alternative approaches to assess the effect of GR activation in vivo (systemic) and ex vivo (ovary culture): here the RNA seq did show again some changes in germ cells and many in the soma.They exclude oocyte-specific GR signaling inhibition via beta isoforms.Perinatal male germline shows differential splicing regulation in response to systemic Dex administration, results were backed up with q-PCR analysis of splicing factors.Weaknesses:COMMENT #1: The presence of a protein cannot be entirely excluded based on IF data

We agree that very low levels of GR could escape the detection by IF and confocal imaging. We feel that our IF data do match transcript data in our validation studies of the GR KO using (1) qRT-PCR on fetal ovary in Fig 2E and (2) scRNA-seq in germ cells and ovarian soma in Fig S2B.

COMMENT #2: (staining of spermatids is referred to but not shown).

You are correct that this statement was based on a morphological identification of spermatids using DAPI morphology. We have performed a co-stain for GR with the spermatocyte marker SYCP3, and the spermatid/spermatozoa marker PNA (Peanut Agglutinin; from Arachis hypogaea) in adult testis tissue. We have updated Figure 4D to reflect this change, as well as the corresponding text in the Results section.

COMMENT #3: The authors do not consider post-transcriptional level (a) modifications also triggered by GR activation (b) non-coding RNAs (not assessed by seq).

We thank the reviewer for raising this very important point about potential post-transcriptional (non-genomic) effects of GR in the fetal oocyte. We agree that while our RNA-seq results show only a minimal transcriptional response, we cannot rule out a non-canonical signaling function of GR, such as the regulation of cellular kinases (as reviewed elsewhere1), or the regulation of non coding RNAs at the post-transcriptional level, and we have amended the discussion to include a sentence on this point. However, while we fully acknowledge the possibility of GR regulating non-genomic level cellular signaling, we chose not to explore this option further based on the lack of any overall functional effect on meiotic progression when GR signaling was perturbed- either by KO (Figure 2D) or dex-mediated activation (Figure S3C).

COMMENT #4: Sequencing techniques used are not total RNA but either are focused on all polyA transcripts (10x) or only assess the 3' prime end and hence are not ideal to study splicing

We thank the reviewer for raising this concern, however this statement is not correct and we have clarified this point in the Results section to explain how the sequencing libraries of the male germ cell RNA-seq were prepared. We agree that certain sequencing techniques (such as 3’ Tag-Seq) that generate sequencing libraries from a limited portion of an entire transcript molecule are not appropriate for analysis of differential splicing. This was not the case, however, for the RNA-seq libraries prepared on our male germ cells treated with dexamethasone. These libraries were constructed using full length transcripts that were reverse transcribed using random hexamer priming, thus accounting for sequencing coverage across the full transcript length. As a result, this type of library prep technique should be sufficient for capturing differential splicing events along the length of the transcript. We do, however, point out that these libraries were constructed on polyA-enriched transcripts. Thus while we obtained full length transcript coverage for these polyA transcripts, any differential splicing taking place in non poly-adenylated RNA moieties were not captured. While we are excited about the possibility of exploring GR-mediated splicing regulation of other RNA species in the future, we chose to focus the scope of our current study on polyA mRNA molecules specifically.

COMMENT #5: The number of replicates in the low input seq is very low and hence this might be underpowered

While the number of replicates (n=3-4 per condition) is sufficient for performing statistical analysis of a standard RNA-seq experiment, we do acknowledge and agree with the reviewer that low numbers of FACS-sorted germ cells from individual embryos combined with the low input 3’ Tag-Seq technique could have led to higher sample variability than desired. Given that we validated our bulk RNA-seq analysis of GR knockout ovaries using an orthogonal single-cell RNA-seq approach, we feel that our conclusions regarding a lack of transcriptional changes upon GR deletion remain valid.

COMMENT #6: Since Dex treatment showed some (modest) changes in oocyte RNA - effects of GR depletion might only become apparent upon Dex treatment as an interaction.

We may be missing the nuance of this point, but our interpretation of an effect that is seen only when the KO is treated with Dex would be that the mechanism would not be autonomous in germ cells but indirect or off-target.

COMMENT #7: Effects in oocytes following systemic Dex might be indirect due to GR activation in the soma.

As both the oocytes and ovarian soma express GR during the window of dex administration, we agree that it is possible that the few modest changes seen in the oocyte transcriptome are the result of indirect effects following robust GR signaling in the somatic compartment. However, given that these modest oocyte transcript changes in response to dex treatment did not significantly alter the ability of oocytes to progress through meiosis, we chose not to explore this mechanism further.

COMMENT #8: Even though ex vivo culture of ovaries shows GR translocation to the nucleus it is not sure whether the in vivo systemic administration does the same.ANDThe conclusion that fetal oocytes are resistant to GR manipulation is very strong, given that "only" poly A sequencing and few replicates of 3-prime sequencing have been analyzed and information is lacking on whether GR is activated in germ cells in the systemically dex-injected animals.

If we understand correctly, the first part refers to a technical limitation and the second part takes issue with our interpretation of the data. For the former, we appreciate this astute insight on the conundrum of detecting a response to systemic dex in fetal oocytes, which is generally monitored by nuclear translocation of GR. As shown in Figure 1A and 1B, GR localization is overwhelmingly nuclear in fetal oocytes of WT animals at E13.5 without addition of any dex. We could not, therefore, use GR translocation as a proxy for activation in response to dex treatment. We instead used ex vivo organ culture to monitor localization changes, as we were able to maintain fetal ovaries ex vivo in hormone-depleted and ligand negative conditions. As shown in Fig. 3, these defined culture conditions elicited a shift of GR to the cytoplasm of fetal oocytes. This led us to conclude that GR is capable of translocating between nucleus and cytoplasm in fetal oocytes, and we were able to counteract this loss in nuclear localization by providing dex ligand in the media.

We feel that our conclusion that oocytes are resistant to manipulation of glucocorticoid signaling despite their possession of the receptor and capacity for nuclear translocation is substantiated by multiple results: meiotic phenotyping, bulk RNA-seq and scRNA-seq analysis of both GR KO and dex dosed mice. Our basis for testing the timing and fidelity of meiotic prophase I was the coincident onset of GR expression in female germ cells at E13, and the disappearance of GR in neonatal oocytes as they enter meiotic arrest. The lack of transcriptional changes observed in oocytes in response to dex has made it even more challenging to demonstrate a bona fide “activation” of GR. Observation of a dose-dependent induction of the canonical GR response gene Fkbp5 in the somatic cells of the fetal ovary (Figure S3A and 3A) affirmed that dex traverses the placenta. We agree with the reviewer that it remains possible that dex or GR KO could lead to changes in epigenetic marks or small RNAs in oocytes, and have mentioned these possibilities in the discussion, but we note that even epigenetic perturbations during oocyte development such as the loss of Tet1 or Dnmt1 result in measurable changes in the transcriptome and the timing of meiotic prophase 2–4.

COMMENT #9: This work is a good reference point for researchers interested in glucocorticoid hormone signaling fertility and RNA splicing. It might spark further studies on germline-specific GR functions and the impact of GR activation on alternative splicing. While the study provides a characterization of GR and some aspects of GR perturbation, and the negative findings in this study do help to rule out a range of specific roles of GR in the germline, there is still a range of other potential unexplored options. The introduction of the study eludes to implications for intergenerational effects via epigenetic modifications in the germline, however, it does not mention that the indirect effects of reproductive tissue GR signaling on the germline have indeed already been described in the context of intergenerational effects of stress.

The reviewer raises an excellent point that we have not made sufficient distinction in our manuscript between prior studies of gestational stress and preconception stress and the light that our work may shed on those findings. We have revised the introduction to clarify this difference, and added reference to an outstanding study that identifies glucocorticoid-induced changes to microRNA cargo of extracellular vesicles shed by epididymal epithelial cells that when transferred to mature sperm can induce changes in the HPA axis and brain of offspring 5. Interestingly, this GR-mediated effect in the epididymal epithelial cells concurs with our observation in the adult testis that GR can be detected only cKit+ spermatogonia but not in subsequent stages of spermatids.

COMMENT #10: Also, the study does not assess epigenetic modifications.

We agree with the reviewer that exploring the role of GR in regulating epigenetic modifications within the germline is an area of extreme interest given the potential links between stress and transgenerational epigenetic inheritance. As this is a broader topic that requires a more thorough and comprehensive set of experiments, we have intentionally chosen to keep this work separate from the current study, and hope to expand upon this topic in the future.

COMMENT #11: The conclusion that the persistence of a phenotype for up to three generations suggests that stress can induce lasting epigenetic changes in the germline is misleading. For the reader who is unfamiliar with the field, it is important to define much more precisely what is referred to as "a phenotype". Furthermore, this statement evokes the impression that the very same epigenetic changes in the germline have been observed across multiple generations.

We see how this may be misleading, and we have amended the text of the introduction and discussion accordingly to avoid the use of the term “phenotype”.

COMMENT #12: The evidence of the presence of GR in the germline is also somewhat limited - since other studies using sequencing have detected GR in the mature oocyte and sperm.

As described above in response to Comment #2, we have included immunostaining of adult testis in a revised Figure 4D and shown that we detect GR in PLZF+ and cKIT+ spermatogonia. We also show low/minimal expression in some (SYCP3+) early meiotic spermatocytes, but not in (Lectin+) spermatids. We are not aware of any studies that have shown expression of GR protein in the mature oocyte.

COMMENT #13: The discussion ends again on the implications of sex-specific differences of GR signaling in the context of stress-induced epigenetic inheritance. It states that the observed differences might relate to the fact that there is more evidence for paternal lineage findings, without considering that maternal lineage studies in epigenetic inheritance are generally less prevalent due to some practical factors - such as more laborious study design making use of cross-fostering or embryo transfer.

We thank the reviewer for this valid point, and we have amended the discussion section.

**Reviewer #2 (Public Review):**
Summary:There is increasing evidence in the literature that rodent models of stress can produce phenotypes that persist through multiple generations. Nevertheless, the mechanism(s) by which stress exposure produces phenotypes are unknown in the directly affected individual as well as in subsequent offspring that did not directly experience stress. Moreover, it has also been shown that glucocorticoid stress hormones can recapitulate the effects of programmed stress. In this manuscript, the authors test the compelling hypothesis that glucocorticoid receptor (GR)-signaling is responsible for the transmission of phenotypes across generations. As a first step, the investigators test for a role of GR in the male and female germline. Using knockouts and GR agonists, they show that although germ cells in male and female mice have GR that appears to localize to the nucleus when stimulated, oocytes are resistant to changes in GR levels. In contrast, the male germline exhibits changes in splicing but no overt changes in fertility.Strengths:Although many of the results in this manuscript are negative, this is a careful and timely study that informs additional work to address mechanisms of transmission of stress phenotypes across generations and suggests a sexually dimorphic response to glucocorticoids in the germline. The work presented here is well-done and rigorous and the discussion of the data is thoughtful. Overall, this is an important contribution to the literature.
**Reviewer #1 (Recommendations For The Authors):**
RECOMMENDATION #1: To assess whether in females the systemic Dex administration directly activates GR in oocytes it would be great to assess GR activation following Dex administration, and ideally to see the effects abolished when Dex is administered to germline-specific KO animals.

In regard to the recommendation to assess GR activation in response to systemic dex administration, we refer the reviewer back to our response in Comment #8 highlighting the difficulties defining and measuring GR activation in the germline.

This therefore has made it difficult to assess whether any of the modest effects seen in response to dex are abolished in our germline-specific KO animals. While repeating our RNA-seq experiment in dex-dosed germline KO animals would address whether the ~60 genes induced in oocytes are the result of oocyte-intrinsic GR activity, we have decided not to explore this mechanism further due to the overall lack of a functional effect on meiotic progression in response to dex (Figure S3C).

RECOMMENDATION #2: To further strengthen the link between GR and alternative splicing it would be great to see the dex administration experiment repeated in germline specific GR KO's.

While we understand the reviewer’s suggestion to explore whether deletion of GR in the spermatogonia is sufficient to abrogate the dex-mediated decreases in splice factor expression, we chose not to explore the details of this mechanism given that deletion of GR in the male germline does not impair fertility (Figure 6).

RECOMMENDATION #3: I am wondering how much a given reduction in one of the splicing factors indeed affects splicing events. Can the authors relate this to literature, or maybe an in vitro experiment can be done to see whether the level of differential splicing events detected is in a range that can be expected in the case of the magnitude of splicing factor reduction?

It has been shown in many instances in the literature that a full genetic deletion of a single splice factor leads to impairments in spermatogenesis, and ultimately infertility 6–16. We suspect that dex treatment leads to fewer differential splicing events than a full splice factor deletion, given that dex treatment causes a broader decrease in splice factor expression without entirely abolishing any single splice factor. We have amended the discussion section to include this point. While we share the reviewer’s curiosity to compare the effects of dex vs genetic deletion of splicing machinery on the overall magnitude of differential splicing events, we unfortunately do not have access to mice with a floxed splice factor at this time. While we have considered knocking out one or more splice factors in an ex vivo cultured testis to compare alongside dex treatment, our efforts to date have proven unsuccessful due to high cell death upon culture of the postnatal testis for more than 24 hours.

RECOMMENDATION #4: It is unclear from the methods whether in germline-specific KO's also the controls received tamoxifen.

We thank the reviewer for catching this missing piece of information. All control embryos that were assessed received an equivalent dose of tamoxifen to the germline-specific KO embryos. The only difference between cKOs and controls was the presence of the Cre transgene. We have updated the Materials and Methods 3’ Tag-Seq sample preparation section to include the sentence: “Both GRcKO/cKO and control GRflox/flox embryos were collected from tamoxifen-injected dams, and thus were equally exposed to tamoxifen in utero”.

**Reviewer #2 (Recommendations For The Authors):**
I just have only a few comments/questions.RECOMMENDATION #5: It is somewhat surprising that GR is expressed in female germ cells, yet there doesn't seem to be a requirement. Is there any indication of what it does? Is the long-term stability of the germline compromised?

We thank the reviewer for these questions, and we agree that it was quite surprising to find a lack of GR function in the female germline despite its robust expression. The question of whether loss of GR affects the long-term stability of the female germline is interesting, given that similar work in GR KO zebrafish has shown impairments to female reproductive capacity, yet only upon aging 17–19.

While we have shared interest in this question, technical limitations thus far have prevented us from properly assessing the effect of GR loss in aged females. Homozygous deletion of GR results in embryonic lethality at approximately E17.5. Conditional deletion of GR using Oct4-CreERT2 with a single dose of tamoxifen (2.5 mg / 20g mouse) at E9.5 results in complete deletion of GR by E10.5, although dams consistently suffer from dystocia and are no longer able to deliver viable pups. While using the more active tamoxifen metabolite (4OHT) at 0.1 mg / 20g has allowed for successful delivery, the resulting deletion rate is very poor (see qPCR results in panel below, left). While using half the dose of standard tamoxifen (1.25 mg / 20g mouse) at E9.5 has on rare occasions led to a successful delivery, the resulting recombination efficiency is insufficient (Author response image 1 right panel).

**Author response image 1. sa2fig1:** 

While a Blimp1-Cre conditional KO model was used to assess male fertility on GR deletion, we believe this model may not be ideal for studying fertility in the context of aging. While Blimp1-Cre is highly specific to the germ cells within the gonad, there are many cell types outside of the gonad that express Blimp1, including the skin and certain cells of the immune system. It is unclear, particularly over the course of aging, whether any effects on fertility seen would be due to an oocyte-intrinsic effect, or the result of GR loss elsewhere in the body. While we hope to explore the role of GR in the aging oocyte further using alternative Cre models in the future, this is currently outside the scope of this work.

RECOMMENDATION #6: Figure 5b: what is the left part of that panel? Is it the same volcano plot for germ cells as shown in part a but with splicing factors?

We apologize if this panel was unclear. Yes, the left panel of Figure 5B is in fact the same volcano plot in 5A, labeled with splicing factors instead of top genes. We have edited Figure 5B and corresponding figure legend to clarify this.

References:

1. Oakley, R.H., and Cidlowski, J.A. (2013). The biology of the glucocorticoid receptor: New signaling mechanisms in health and disease. J. Allergy Clin. Immunol. 132, 1033–1044. 10.1016/j.jaci.2013.09.007.

2. Hargan-Calvopina, J., Taylor, S., Cook, H., Hu, Z., Lee, S.A., Yen, M.-R., Chiang, Y.-S., Chen, P.-Y., and Clark, A.T. (2016). Stage-Specific Demethylation in Primordial Germ Cells Safeguards against Precocious Differentiation. Dev. Cell 39, 75–86. 10.1016/j.devcel.2016.07.019.

3. Hill, P.W.S., Leitch, H.G., Requena, C.E., Sun, Z., Amouroux, R., Roman-Trufero, M., Borkowska, M., Terragni, J., Vaisvila, R., Linnett, S., et al. (2018). Epigenetic reprogramming enables the transition from primordial germ cell to gonocyte. Nature 555, 392–396. 10.1038/nature25964.

4. Eymery, A., Liu, Z., Ozonov, E.A., Stadler, M.B., and Peters, A.H.F.M. (2016). The methyltransferase Setdb1 is essential for meiosis and mitosis in mouse oocytes and early embryos. Development 143, 2767–2779. 10.1242/dev.132746.

5. Chan, J.C., Morgan, C.P., Leu, N.A., Shetty, A., Cisse, Y.M., Nugent, B.M., Morrison, K.E., Jašarević, E., Huang, W., Kanyuch, N., et al. (2020). Reproductive tract extracellular vesicles are sufficient to transmit intergenerational stress and program neurodevelopment. Nat Commun 11, 1499. 10.1038/s41467-020-15305-w.

6. Kuroda, M., Sok, J., Webb, L., Baechtold, H., Urano, F., Yin, Y., Chung, P., Rooij, D.G. de, Akhmedov, A., Ashley, T., et al. (2000). Male sterility and enhanced radiation sensitivity in TLS−/− mice. Embo J 19, 453–462. 10.1093/emboj/19.3.453.

7. Liu, W., Wang, F., Xu, Q., Shi, J., Zhang, X., Lu, X., Zhao, Z.-A., Gao, Z., Ma, H., Duan, E., et al. (2017). BCAS2 is involved in alternative mRNA splicing in spermatogonia and the transition to meiosis. Nat Commun 8, 14182. 10.1038/ncomms14182.

8. Li, H., Watford, W., Li, C., Parmelee, A., Bryant, M.A., Deng, C., O’Shea, J., and Lee, S.B. (2007). Ewing sarcoma gene EWS is essential for meiosis and B lymphocyte development. J Clin Invest 117, 1314–1323. 10.1172/jci31222.

9. O’Bryan, M.K., Clark, B.J., McLaughlin, E.A., D’Sylva, R.J., O’Donnell, L., Wilce, J.A., Sutherland, J., O’Connor, A.E., Whittle, B., Goodnow, C.C., et al. (2013). RBM5 Is a Male Germ Cell Splicing Factor and Is Required for Spermatid Differentiation and Male Fertility. Plos Genet 9, e1003628. 10.1371/journal.pgen.1003628.

10. Zagore, L.L., Grabinski, S.E., Sweet, T.J., Hannigan, M.M., Sramkoski, R.M., Li, Q., and Licatalosi, D.D. (2015). RNA Binding Protein Ptbp2 Is Essential for Male Germ Cell Development. Mol Cell Biol 35, 4030–4042. 10.1128/mcb.00676-15.

11. Xu, K., Yang, Y., Feng, G.-H., Sun, B.-F., Chen, J.-Q., Li, Y.-F., Chen, Y.-S., Zhang, X.-X., Wang, C.-X., Jiang, L.-Y., et al. (2017). Mettl3-mediated m6A regulates spermatogonial differentiation and meiosis initiation. Cell Res 27, 1100–1114. 10.1038/cr.2017.100.

12. Horiuchi, K., Perez-Cerezales, S., Papasaikas, P., Ramos-Ibeas, P., López-Cardona, A.P., Laguna-Barraza, R., Balvís, N.F., Pericuesta, E., Fernández-González, R., Planells, B., et al. (2018). Impaired Spermatogenesis, Muscle, and Erythrocyte Function in U12 Intron Splicing-Defective Zrsr1 Mutant Mice. Cell Reports 23, 143–155. 10.1016/j.celrep.2018.03.028.

13. Ehrmann, I., Crichton, J.H., Gazzara, M.R., James, K., Liu, Y., Grellscheid, S.N., Curk, T., Rooij, D. de, Steyn, J.S., Cockell, S., et al. (2019). An ancient germ cell-specific RNA-binding protein protects the germline from cryptic splice site poisoning. Elife 8, e39304. 10.7554/elife.39304.

14. Legrand, J.M.D., Chan, A.-L., La, H.M., Rossello, F.J., Änkö, M.-L., Fuller-Pace, F.V., and Hobbs, R.M. (2019). DDX5 plays essential transcriptional and post-transcriptional roles in the maintenance and function of spermatogonia. Nat Commun 10, 2278. 10.1038/s41467-019-09972-7.

15. Yuan, S., Feng, S., Li, J., Wen, H., Liu, K., Gui, Y., Wen, Y., and Wang, X. (2021). hnRNPH1 recruits PTBP2 and SRSF3 to cooperatively modulate alternative pre-mRNA splicing in germ cells and is essential for spermatogenesis and oogenesis. 10.21203/rs.3.rs-1060705/v1.

16. Wu, R., Zhan, J., Zheng, B., Chen, Z., Li, J., Li, C., Liu, R., Zhang, X., Huang, X., and Luo, M. (2021). SYMPK Is Required for Meiosis and Involved in Alternative Splicing in Male Germ Cells. Frontiers Cell Dev Biology 9, 715733. 10.3389/fcell.2021.715733.

17. Maradonna, F., Gioacchini, G., Notarstefano, V., Fontana, C.M., Citton, F., Valle, L.D., Giorgini, E., and Carnevali, O. (2020). Knockout of the Glucocorticoid Receptor Impairs Reproduction in Female Zebrafish. Int J Mol Sci 21, 9073. 10.3390/ijms21239073.

18. Facchinello, N., Skobo, T., Meneghetti, G., Colletti, E., Dinarello, A., Tiso, N., Costa, R., Gioacchini, G., Carnevali, O., Argenton, F., et al. (2017). nr3c1 null mutant zebrafish are viable and reveal DNA-binding-independent activities of the glucocorticoid receptor. Sci Rep-uk 7, 4371. 10.1038/s41598-017-04535-6.

19. Faught, E., Santos, H.B., and Vijayan, M.M. (2020). Loss of the glucocorticoid receptor causes accelerated ovarian ageing in zebrafish. Proc Royal Soc B 287, 20202190. 10.1098/rspb.2020.2190.